# Unsupervised Anomaly Detection by Robust Collaborative Autoencoders

## Abstract

Unsupervised anomaly detection plays a critical role in many real-world applications, from computer security to healthcare. A common approach based on deep learning is to apply autoencoders to learn a feature representation of the normal (non-anomalous) observations and use the reconstruction error of each observation to detect anomalies present in the data. However, due to the high complexity brought upon by over-parameterization of the deep neural networks (DNNs), the anomalies themselves may have small reconstruction errors, which degrades the performance of these methods. To address this problem, we present a robust framework for detecting anomalies using collaborative autoencoders. Unlike previous methods, our framework does not require supervised label information nor access to clean (uncorrupted) examples during training. We investigate the theoretical properties of our framework and perform extensive experiments to compare its performance against other DNN-based methods. Our experimental results show the superior performance of the proposed framework as well as its robustness to noise due to missing value imputation compared to the baseline methods.

## 1 Introduction

Anomaly detection (AD) is the task of identifying abnormal observations in the data. It has been successfully applied to many applications, from malware detection to medical diagnosis (Chandola et al., 2009). Driven by the success of deep learning, AD methods based on deep neural networks (DNNs) (Zhou & Paffenroth, 2017; Aggarwal & Sathe, 2017; Ruff et al., 2018; Zong et al., 2018; Hendrycks et al., 2018) have attracted increasing attention recently. Unfortunately, DNN methods have several known drawbacks when applied to AD problems. First, since many of them are based on the supervised learning approach (Hendrycks et al., 2018), this requires labeled examples of anomalies, which are often expensive to acquire and may not be representative enough in non-stationary environments. Supervised AD methods are also susceptible to the class imbalance problem as anomalies are rare compared to normal observations. Some DNN methods rely on having access to clean data to ensure that the feature representation learning is not contaminated by anomalies during training (Zong et al., 2018; Ruff et al., 2018; Pidhorskyi et al., 2018; Fan et al., 2020). This limits their applicability as acquiring a representative clean data itself is a tricky problem. Due to these limitations, there have been concerted efforts to develop robust unsupervised DNN methods that do not assume the availability of supervised labels nor clean training data (Chandola et al., 2009; Liu et al., 2019). Deep autoencoders are perhaps one of the most widely used unsupervised AD methods (Sakurada & Yairi, 2014; Vincent et al., 2010). An autoencoder compresses the original data by learning a latent representation that minimizes the reconstruction loss. It is based on the working assumption that normal observations are easier to compress than anomalies. Unfortunately, such an assumption may not hold in practice since DNNs are often over-parameterized and have the capability to overfit the anomalies (Zhang et al., 2016), thus degrading their overall performance.

To improve their performance, the unsupervised DNN methods must consider the trade-off between model capacity and overfitting to the anomalies. One way to control the model capacity is through regularization. Many regularization methods for deep networks have been developed to control model capacity, e.g., by constraining the norms of the model parameters or explicitly perturbing the training process (Srivastava et al., 2014). However, these approaches do not prevent the networks from being able to perfectly fit random data (Zhang et al., 2016). As a consequence, the regulariza-

tion approaches cannot prevent the anomalies from being *memorized*, especially in an unsupervised learning setting.

Our work is motivated by recent advances in supervised learning on the robustness of DNNs for noisy labeled data by learning the weights of the training examples (Jiang et al., 2017; Han et al., 2018). Unlike previous studies, our goal is to learn the weights in an unsupervised learning fashion so that normal observations are assigned higher weights than the anomalies when calculating reconstruction error. The weights help to reduce the influence of anomalies when learning a feature representation of the data. Since existing approaches for weight learning are supervised, they are inapplicable to unsupervised AD. Instead, we propose an unsupervised robust collaborative autoencoders (RCA) method that trains a pair of autoencoders in a collaborative fashion and jointly learns their model parameters and sample weights. Each autoencoder selects a subset of samples with lowest reconstruction errors from a mini-batch to learn their feature representation. By discarding samples with high reconstruction errors, the algorithm is biased towards learning the representation for clean data, thereby reducing its risk of memorizing anomalies. However, by selecting only easy-to-fit samples in each iteration, this may lead to premature convergence of the algorithm without sufficient exploration of the loss surface. Thus, instead of selecting the samples to update its own model parameters, each autoencoder will shuffle its selected samples to the other autoencoder, who will use the samples to update their model parameters. The sample selection procedure is illustrated in Figure 1. During the testing phase, we apply the dropout mechanism used in training to produce multiple output predictions for each test point by repeating the forward pass multiple times. These ensemble of outputs are then aggregated to obtain a more robust estimate of the anomaly score.

The main contributions of this paper are as follows. First, we present a novel framework for unsupervised AD using robust collaborative autoencoders (RCA). Second, we provide rigorous theoretical analysis to understand the mechanism behind RCA. We also describe the convergence of RCA to the solution obtained if it was trained on clean data only. We show that the worst-case scenario for RCA is better than conventional autoencoders and analyze the conditions under which RCA is guaranteed to find the anomalies. Finally, we empirically demonstrate that RCA outperforms state-of-the-art unsupervised AD methods for the majority of the datasets used in this study, even in the presence of noise due to missing value imputation.

## 2 RELATED WORK

There are numerous methods developed for anomaly detection, a survey of which can be found in Chandola et al. (2009). Reconstruction-based methods, such as principal component analysis (PCA) and autoencoders, are popular approaches, whereby the input data is projected to a lower-dimensional space before it was transformed back to its original feature space. The distance between the input and reconstructed data is used to determine the anomaly scores of the data points. More advanced unsupervised AD methods have been developed recently. Zhou & Paffenroth (2017) combined robust PCA with an autoencoder to decompose the data into a mixture of normal and anomaly parts. Zong et al. (2018) jointly learned a low dimensional embedding and density of the data, using the density of each point as its anomaly score while Ruff et al. (2018) extended the traditional one-class SVM approach to a deep learning setting. Wang et al. (2019) applied an end-to-end self-supervised learning approach to the unsupervised AD problem. However, their approach is designed for image data, requiring operations such as rotation and patch reshuffling.

Despite the recent progress on deep unsupervised AD, current methods do not explicitly prevent the neural network from incorporating anomalies into their learned representation, thereby degrading the model performance. One way to address the issue is by assigning a weight to each data point, giving higher weights to the normal data to make the model more robust against anomalies. The idea of learning a weight for each data point is not new in supervised learning. A classic example is boosting (Freund et al., 1996), where hard to classify examples are assigned higher weights to encourage the model to classify them more accurately. An opposite strategy is used in self-paced learning (Kumar et al., 2010), where the algorithm assigns higher weights to easier-to-classify examples and lower weights to harder ones. This strategy was also used by other methods for learning from noisy labeled data, including Jiang et al. (2017) and Han et al. (2018). Furthermore, there are many studies providing theoretical analysis on the benefits of choosing samples with smaller loss to drive the optimization algorithm (Shen & Sanghavi, 2018; Shah et al., 2020).

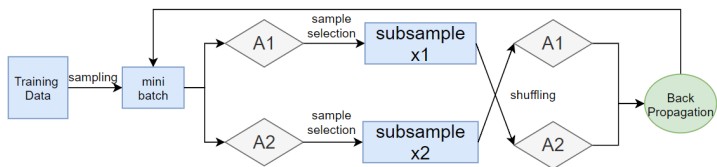

Figure 1: An illustration of the training phase for the proposed RCA framework.

## 3 METHODOLOGY

This section introduces the proposed robust collaborative autoencoder (RCA) framework and analyze its properties. Let $\mathbf{X} \in \mathbb{R}^{n \times d}$ denote the input data, where $n$ is the number of observations and $d$ is the number of features. Our goal is to classify each data point $x_i \in \mathbf{X}$ as an anomaly or a normal observation. Let $\mathcal{O} \subset \mathbf{X}$ denote the set of true anomalies in the data. We assume the anomaly ratio, $\epsilon = |\mathcal{O}|/n$, is given[1] or can be approximately estimated.

The RCA framework trains a pair of autoencoders, $\mathcal{A}_1$ and $\mathcal{A}_2$, with different initializations. In each iteration during training, the autoencoders will each apply a forward pass on a mini-batch randomly sampled from the training data and compute the reconstruction error of each data point in the mini-batch. The data points in the mini-batch are then sorted according to their reconstruction errors and each mini-batch selects the points with lowest errors to be exchanged with the other autoencoder. Each autoencoder subsequently performs a back-propagation step to update its model parameters using the samples it receives from the other autoencoder. Upon convergence, the averaged reconstruction error of each data point is used to determine the anomaly score. A pseudocode of the training phase for RCA is given in Algorithm 1, while the testing phase is given in Algorithm 2.

---

**Algorithm 1:** Robust Collaborative Autoencoders (Training Phase)

---

**input:** training data $\mathbf{X}_{trn}$, test data $\mathbf{X}_{tst}$, reconstruction loss function $\mathcal{L}$, anomaly ratio $\epsilon$, dropout rate $r > 0$, and maximum training epochs: $max\_epoch$;
**return** trained autoencoders, $\mathcal{A}_1^*$ and $\mathcal{A}_2^*$ ;
**Initialize** autoencoders $\mathcal{A}_1$ and $\mathcal{A}_2$; sample selection rate $\beta = 1$ and best loss $\xi^* = +\infty$;
**while** epoch $\leq$ max_epoch **do**
 **for** minibatch $\mathbf{S}$ in $\mathbf{X}_{trn}$ **do**
  $\hat{\mathbf{S}}_1 \leftarrow$ forward$(\mathcal{A}_1, \mathbf{S}, dropout = 0)$, $\hat{\mathbf{S}}_2 \leftarrow$ forward$(\mathcal{A}_2, \mathbf{S}, dropout = 0)$;
  $\mathbf{c}_1 \leftarrow$ sample_selection$(\mathcal{L}(\hat{\mathbf{S}}_1, \mathbf{S}), \beta)$, $\mathbf{c}_2 \leftarrow$ sample_selection$(\mathcal{L}(\hat{\mathbf{S}}_2, \mathbf{S}), \beta)$ ;
  $\hat{\mathbf{S}}_1 \leftarrow$ forward$(\mathcal{A}_1, \mathbf{S}[\mathbf{c}_2], dropout = r)$, $\hat{\mathbf{S}}_2 \leftarrow$ forward$(\mathcal{A}_2, \mathbf{S}[\mathbf{c}_1], dropout = r)$;
  $\mathcal{A}_1 \leftarrow$ backprop$(\hat{\mathbf{S}}_1, \mathbf{S}[\mathbf{c}_2], dropout = r)$, $\mathcal{A}_2 \leftarrow$ backprop$(\hat{\mathbf{S}}_2, \mathbf{S}[\mathbf{c}_1], dropout = r)$ ;
 **end**
 $\hat{\mathbf{X}}_{tst1} \leftarrow$ forward$(\mathcal{A}_1, \mathbf{X}_{tst}, dropout = 0)$, $\hat{\mathbf{X}}_{tst2} \leftarrow$ forward$(\mathcal{A}_2, \mathbf{X}_{tst}, dropout = 0)$;
 $\xi_{test} \leftarrow \mathcal{L}(\hat{\mathbf{X}}_{tst1}, \mathbf{X}_{tst}) + \mathcal{L}(\hat{\mathbf{X}}_{tst2}, \mathbf{X}_{tst})$;
 **if** $\xi_{test} < \xi^*$ **then** $\xi^* = \xi_{test}, \mathcal{A}_1^* = \mathcal{A}_1, \mathcal{A}_2^* = \mathcal{A}_2$ **end**;
 $\beta = \max(\beta - \dfrac{\epsilon}{\text{max\_epoch}}, 1 - \epsilon)$
**end**

---

**Algorithm 2:** Robust Collaborative Autoencoders (Testing Phase)

---

**input:** test data $\mathbf{X}_{tst}$, trained autoencoders $\mathcal{A}_1^*, \mathcal{A}_2^*$, dropout rate $r > 0$, size of ensemble $v$;
**return** anomaly_score ;
Initialize an empty set of reconstruction errors: $\xi = \{\}$.
**for** $i = 1$ **to** $v$ **do**
 $\xi_1 =$ forward$(\mathcal{A}_1^*, \mathbf{X}_{tst}, dropout = r)$, $\xi_2 =$ forward$(\mathcal{A}_2^*, \mathbf{X}_{tst}, dropout = r)$ ;
 $\xi = \xi \cup (\xi_1 + \xi_2)/2$
**end**
$anomaly\_score =$ average$(\xi)$;

---

RCA differs from conventional autoencoders in several ways. First, its autoencoders are trained using only selected data points with small reconstruction errors. The selected points are then exchanged between the autoencoders to avoid premature convergence. Furthermore, in the testing phase, each autoencoder applies a dropout mechanism to generate multiple predicted outputs. The averaged ensemble output is used as the final anomaly score. Details of these steps are given next.

---

[1]In practice, users would typically specify the top-$k$ anomalies to be examined and verified, where $k = n\epsilon$.

### 3.1 SAMPLE SELECTION

Given a mini-batch, $\mathbf{X}_m \subset \mathbf{X}$, our sample selection procedure chooses a subset of points as "clean" samples to update the parameters of an autoencoder by minimizing the following objective function:

$$\min_{\mathbf{w},\mathbf{c}} \sum_{\mathbf{x}_i \in \mathbf{X}_m} c_i f(\mathbf{x}_i, \mathbf{w}), \quad \text{s.t.} \quad \forall i : c_i \in \{0,1\}, \; \mathbf{c}^T \mathbf{1} = \beta n, \tag{1}$$

where $f(\mathbf{x}, \mathbf{w})$ denotes the reconstruction loss of a data point $\mathbf{x}$ for an autoencoder with parameter $\mathbf{w}$. Although $c_i$ is binary-valued, as will be shown later, the probability that a data point is selected to update an autoencoder depends on the probability it is chosen to be part of the mini-batch and the probability it has among the lowest reconstruction errors within the mini-batch.

We use an alternating minimization approach to solve the objective function. By fixing $\mathbf{c}$ and optimizing $\mathbf{w}$, this reduces to solving the standard autoencoder problem using the selected "clean" samples (i.e., those with $c_i = 1$) by applying an optimizer such as Adam (Kingma & Ba, 2014). When $\mathbf{w}$ is fixed, the objective function reduces to a linear programming problem, which admits a simple solution, where the data points are sorted based on their reconstruction loss, $f(\mathbf{x}_i, \mathbf{w})$. We set $c_i = 1$ to $(\beta \times 100)\%$ of the points in the mini-batch with lowest reconstruction loss. This procedure is denoted by the sample_selection$(\cdot)$ function in Algorithm 1. A weight decay approach is applied to $\beta$ when selecting the "clean" samples. In the early stages of training, all the samples in the mini-batch are selected to update the model, with gradually fewer points selected as training progresses. The rationale for this approach is that we should not drop too many data points early, especially in the first few epochs, when the autoencoders have not properly learn the feature representation. The autoencoders start to overfit the anomalies when the number of training epochs is sufficiently large. A linear decay function from $\beta = 1$ to $\beta = 1 - \epsilon$ (see the second last line in Algorithm 1) was found to work well in practice, so we use this setting in our experiments.

Next, we analyze the convergence properties of our sample selection procedure. Let $k$ be the mini-batch size and $\mathbf{w}$ be the current parameter of the autoencoder. Our algorithm selects $(\beta \times 100)\%$ of the data points with lowest errors in the mini-batch for updating the autoencoder. Let $\mathbf{x}_{(i)} \in \mathbf{X}$ denote the data point with $i^{th}$ smallest reconstruction loss among all $n$ points and $p_i(\mathbf{w})$ be the probability that $\mathbf{x}_{(i)}$ is chosen by the sample selection procedure to update the parameters of the autoencoder. Assuming sampling without replacement, we consider two cases: $i \leq \beta k$ and $i > \beta k$. In the first case (when $i \leq \beta k$), $\mathbf{x}_{(i)}$ is used to update the autoencoder as long as it is selected to be part of the mini-batch. In the second case (when $i > \beta k$), $\mathbf{x}_{(i)}$ is chosen only if it is part of the mini-batch and has among the $(\beta k)$-th lowest errors among all data points in the mini-batch. Thus:

$$p_i(\mathbf{w}) = \begin{cases} \frac{\binom{n-1}{k-1}}{\binom{n}{k}} = \frac{k}{n} & \text{if } i \leq \beta k, \\ \frac{\sum_{j=0}^{\beta k - 1} \binom{i-1}{j}\binom{n-i}{k-j-1}}{\binom{n}{k}} & \text{otherwise.} \end{cases} \tag{2}$$

The corresponding probability $p_i(\mathbf{w})$ for sampling with replacement is also provided in the Appendix section. The objective function for our sample selection procedure (Equation 1) can be stated as $\hat{F}(\mathbf{w}) = \sum_{\mathbf{x}_{(i)} \in \mathbf{X}} p_i(\mathbf{w}) f(\mathbf{x}_{(i)}, \mathbf{w})$. Let $\Omega(\mathbf{w}_{sr}^*)$ be the set of stationary points of $\hat{F}(\mathbf{w})$. Furthermore, let $\Omega(\mathbf{w}^*)$ be the set of stationary points for the loss on clean data only, i.e., $F(\mathbf{w}) = \sum_{\mathbf{x}_i \notin \mathcal{O}} f_i(\mathbf{w})$, while $\Omega(\mathbf{w}_{ns}^*)$ be the set of stationary points for the loss on the entire data, i.e., $\sum_{\mathbf{x}_i \in \mathbf{X}} f_i(\mathbf{w})$. For brevity, we have used $f_i(\mathbf{w})$ to denote $f(\mathbf{x}_i, \mathbf{w})$. Furthermore, we denote $\Omega_i(\mathbf{w}^*)$ as the set of stationary points for the individual loss, $f_i(\mathbf{w})$. Our analysis on the convergence properties of our sample selection approach is based on the following assumptions:

**Assumption 1 (Gradient Regularity)** $\max_{i,\mathbf{w}} \|\nabla f_i(\mathbf{w})\| \leq G$.

**Assumption 2 (Bounded Clean Objective)** *Let* $F(\mathbf{w}) = \sum_{\mathbf{x}_i \notin \mathcal{O}} f_i(\mathbf{w})$, *There exists a constant* $B > 0$ *such that the following inequality holds:* $\max_{i,j} |(F(\mathbf{w}_i) - F(\mathbf{w}_j))| \leq B$.

**Assumption 3 (Individual L-smooth)** *For every individual loss* $f_t(\mathbf{w})$, *the following inequality holds:* $\forall i, j : \|\nabla f_t(\mathbf{w}_i) - \nabla f_t(\mathbf{w}_j)\| \leq L_i \|\mathbf{w}_i - \mathbf{w}_j\|$.

**Assumption 4 (Equal Minima)** *The minimum value of every individual loss is the same, i.e.,* $\forall i, j : \min_{\mathbf{w}} f_i(\mathbf{w}) = \min_{\mathbf{w}} f_j(\mathbf{w})$.

**Assumption 5 (Individual Strong Convexity)** *For every individual loss $f_t(\mathbf{w})$, the following inequality holds: $\forall i, j : \|\nabla f_t(\mathbf{w}_i) - \nabla f_t(\mathbf{w}_j)\| \geq \mu_i \|\mathbf{w}_i - \mathbf{w}_j\|$.*

To simplify the notation, we define $L_{\max} = \max_i(L_i)$, $L_{\min} = \min_i(L_i)$, $\mu_{\max} = \max_i(\mu_i)$, and $\mu_{\min} = \min_i(\mu_i)$. Since $F(\mathbf{w})$ is the sum over the loss for clean data, it is easy to see that Assumption 3 implies $F(\mathbf{w})$ is $n(1 - \epsilon)L_{\max}$ smoothness, while Assumption 5 implies that $F(\mathbf{w})$ is $n(1 - \epsilon)\mu_{\min}$ convex. We thus define $M = n(1 - \epsilon)L_{\max}$, and $m = n(1 - \epsilon)\mu_{\min}$.

Note that Assumptions 1-3 are common for non-convex optimization. Assumption 4 is reasonable in an over-parameterized DNN setting (Zhang et al., 2016). Although Assumption 5 is the strongest assumption, it is only used to discuss correctness of our algorithm in Theorem 3, but not for Theorems 1 and 2. A similar assumption has been used in Shah et al. (2020) in their proof of correctness.

We define the constants $\delta > 0$ and $\phi \geq 1$ as follows:

$$\forall \mathbf{x}_i \notin \mathcal{O}, \ \forall \mathbf{x}_j \in \mathcal{O} : \ \max_{\mathbf{v} \in \Omega_i(\mathbf{w}^*), \mathbf{y} \in \Omega(\mathbf{w}^*)} \|\mathbf{v} - \mathbf{y}\| \leq \delta \leq \min_{\mathbf{z} \in \Omega_j(\mathbf{w}^*), \mathbf{y} \in \Omega(\mathbf{w}^*)} \|\mathbf{z} - \mathbf{y}\|,$$
$$\forall \mathbf{x}_j \in \mathcal{O} : \qquad\qquad \max_{\mathbf{z} \in \Omega_j(\mathbf{w}^*), \mathbf{y} \in \Omega(\mathbf{w}^*)} \|\mathbf{z} - \mathbf{y}\| \leq \phi\delta \qquad\qquad (3)$$

If the loss is convex, then $\Omega_i(\mathbf{w}^*) = \{w_i^*\}$ and $\Omega(\mathbf{w}^*) = \{\mathbf{w}^*\}$. The above equation can be simplified to: $\|\mathbf{w}_i^* - \mathbf{w}^*\| \leq \delta \leq \|\mathbf{w}_j^* - \mathbf{w}^*\| \leq \phi\delta, \quad \forall \mathbf{x}_i \notin \mathcal{O}, \ \forall \mathbf{x}_j \in \mathcal{O}$. The constants $\delta$ and $\phi$ thus provide bounds on the distance between $\mathbf{w}_j^*$ of an anomaly point and $\mathbf{w}^*$ for clean data.

We first consider a non-convex setting. Our goal is to determine whether the parameters learned from the samples chosen by our procedure, which optimizes $\hat{F}(\mathbf{w})$, converges to $\mathbf{w}^*$, the solution obtained by minimizing the loss for clean data, $F(\mathbf{w}) = \sum_{\mathbf{x}_i \notin \mathcal{O}} f_i(\mathbf{w})$.

**Theorem 1** *Let $F(\mathbf{w}) = \sum_{\mathbf{x}_i \notin \mathcal{O}} f_i(\mathbf{w})$ be a twice-differentiable function. Consider the sequence $\{\mathbf{w}^{(t)}\}$ generated by optimizing $\hat{F}(\mathbf{w}) = \sum_i p_i(\mathbf{w})f(\mathbf{x}_{(i)}, \mathbf{w})$, i.e., $\mathbf{w}^{(t+1)} = \mathbf{w}^{(t)} - \eta^{(t)} \sum_i p_i(\mathbf{w}^{(t)})\nabla f(\mathbf{x}_{(i)})$. Let $\max_{\mathbf{w}^{(t)}} \|\nabla F(\mathbf{w}^{(t)}) - \sum_i p_i(\mathbf{w}^{(t)})\nabla f_{(i)}(\mathbf{w}^{(t)})\|^2 = C$. Based on Assumptions 1-3, if $\eta^{(t)}$ satisfies $\sum_{t=1}^{\infty} \eta^{(t)} = \infty$ and $\sum_{t=1}^{\infty} \eta^{(t)^2} \leq \infty$, then $\min_{t=0,1,\cdots,T} \|\nabla F(\mathbf{w}^{(t)})\|^2 \to C$ when $T \to \infty$.*

**Remark 1** *The preceding theorem shows the convergence property of optimizing $\hat{F}(\mathbf{w})$ to a $C$-approximated stationary point of the loss function for clean data, where $C$ depends on the sample selection approach. For example, if $p_i(\mathbf{w}) = 1, \forall \mathbf{x}_i \notin \mathcal{O}$ and $p_i(\mathbf{w}) = 0, \ \forall \mathbf{x}_i \in \mathcal{O}$, then $C = 0$. In this ideal situation, the solution for optimizing $\hat{F}(\mathbf{w})$ reduces to that for vanilla SGD on clean data.*

Next, we compare the stationary point of $\hat{F}(\mathbf{w})$ against the stationary point of the loss function for the entire data (without sample selection).

**Theorem 2** *Let $F(\mathbf{w}) = \sum_{\mathbf{x}_i \notin \mathcal{O}} f_i(\mathbf{w})$ be a twice-differentiable function and $C$ is defined in Theorem 1. Consider the sequence $\{\mathbf{w}_{sr}\}$ generated by optimizing $\hat{F}(\mathbf{w}) = \sum_i p_i(\mathbf{w})f(\mathbf{x}_{(i)}, \mathbf{w})$, i.e., $\mathbf{w}^{(t+1)} = \mathbf{w}^{(t)} - \eta^{(t)} \sum_i p_i(\mathbf{w}^{(t)})\nabla f(\mathbf{x}_{(i)})$ and the sequence $\{\mathbf{w}_{ns}\}$ generated by standard SGD on the entire data, $\mathbf{w}^{(t+1)} = \mathbf{w}^{(t)} - \eta^{(t)}\nabla f(\mathbf{x}_i)$. Based on Assumptions 1- 3 and $C \leq (\min(n\epsilon G, M\delta))^2$, if $\eta^{(t)}$ satisfies $\sum_{t=1}^{\infty} \eta^{(t)} = \infty$ and $\sum_{t=1}^{\infty} \eta^{(t)^2} \leq \infty$, then there exists a large enough $T$ and $\tilde{\mathbf{w}} \in \Omega(\mathbf{w}_{ns}^*)$ such that $\min_{t=0,1,...,T} \|\nabla F(\mathbf{w}_{sr}^{(t)})\| \leq \|\nabla F(\tilde{\mathbf{w}})\|$.*

**Remark 2** *This theorem is analogous to the result given in Shah et al. (2020), which has a convex assumption on their loss function, whereas our theorem is applicable even for the non-convex case. Although the theorem is for worst case analysis, our experiments show that, on average, our method easily outperforms other DNN methods that use all the data.*

Theorem 2 suggests that, as long as $C$ is smaller than a threshold, sample selection gives a better convergence to the stationary points for clean data, compared to using all the data. As the anomaly ratio increases or the distance to nearest outlier increases, sample selection will improve the convergence to stationary points for clean data in the worst case scenario compared to no sample selection.

Below we give a sufficient condition for guaranteeing correctness when the objective is restricted to a convex setting. We assume that $\forall \mathbf{x}_i \notin \mathcal{O} : f_i(\mathbf{w}^*) = 0$ and $\forall \mathbf{x}_j \in \mathcal{O} : f_j(\mathbf{w}^*) > 0$. Assuming $f(\mathbf{w})$ is convex and its gradient is upper bounded, there exists a ball of radius $r > 0$ around $w^*$

defined as follows:

$$\mathcal{B}_r\left(\boldsymbol{w}^*\right) = \left\{\boldsymbol{w} \mid f_i(\boldsymbol{w}) < f_j(\boldsymbol{w}), \forall \mathbf{x}_i \notin \mathbb{O}, \mathbf{x}_j \in \mathbb{O}, \|\boldsymbol{w} - \boldsymbol{w}^*\| \leq r\right\}.$$

The above definition describes a ball around the optimal point, in which the normal observations have a smaller loss than the anomalies. Based on this definition, the following theorem describes the sufficient condition for our algorithm to converge to a solution within the ball.

**Theorem 3** *Let* $F(\mathbf{w}) = \sum_{\mathbf{x}_i \notin \mathcal{O}} f_i(\mathbf{w})$ *be a twice-differentiable function and* $L_{max}^c = \max_{\mathbf{x}_i \notin \mathcal{O}}(L_i)$ *be the maximum Lipschitz smoothness for the clean data and* $\mu_{min}^o = \min_{\mathbf{x}_j \in \mathcal{O}}(\mu_j)$ *be the minimum convexity for anomalies. Consider the sequence* $\{\mathbf{w}_{sr}\}$ *generated by* $\mathbf{w}^{(t+1)} = \mathbf{w}^{(t)} - \eta^{(t)} \sum_i p_i(\mathbf{w}^{(t)}) \nabla f(\mathbf{x}_i)$ *and* $\max_{\mathbf{w}^{(t)}} \|\nabla F(\mathbf{w}^{(t)}) - \sum_i p_i(\mathbf{w}^{(t)}) \nabla f_i(\mathbf{w}^{(t)})\|^2 = C$. *Define* $\kappa = \sqrt{\frac{L_{max}^c}{\mu_{min}^o}}$ *and suppose Assumptions 1-5 hold. If* $\eta^{(t)}$ *satisfy* $\sum_{t=1}^{\infty} \eta^{(t)} = \infty$, $\sum_{t=1}^{\infty} \eta^{(t)^2} \leq \infty$, *and* $C \leq \left(\frac{\delta}{(1+\kappa)m}\right)^2 = \mathbb{O}\left(\frac{\delta}{\kappa}\right)^2$, *then there exists* $r > 0$ *such that* $\mathbf{w}_{sr}^* \in \mathcal{B}_r\left(\boldsymbol{w}^*\right)$.

The proof is given in the Appendix section. The convergence guarantee depends on having a small enough value of $C$, which is related to $\delta$, the distance between the nearest anomalies and the normal points, and the landscape of the loss surface $\kappa$. A small $\kappa$ suggests that the loss surface is very sharp for anomalies (large $\mu_{min}^o$) but flat for normal data (small $L_{max}^c$). In this case, most areas of the loss surface will have a smaller loss for normal observations but larger loss for anomalies (assuming equal minima among all the points). Due to their larger loss, the anomalies have smaller probability to be selected as "clean" sample by the proposed RCA algorithm.

The analysis above shows that sample selection benefits convergence of our method to the stationary points for clean data. However, our ultimate goal is to improve generalization performance, not just converging to good stationary points of the training data. When sample selection is applied to just one autoencoder, the algorithm may converge too quickly as we use only samples with low reconstruction loss to compute the gradient, making it susceptible to overfitting (Zhang et al., 2016). Thus, instead of using only the self-selected samples for model update, we train two autoencoders collaboratively and shuffle the selected samples between them to avoid overfitting. Similar ideas have been found to be effective in supervised learning for data with noisy labels (Han et al., 2018).

### 3.2 ENSEMBLE EVALUATION

Unsupervised anomaly detection using an ensemble of model outputs have been shown to be highly effective in previous studies (Liu et al., 2008; Zhao et al., 2019; Emmott et al., 2015; Aggarwal & Sathe, 2017). However, incorporating ensemble method to deep learning is a challenging problem as it is expensive to train a large number of DNNs. In this paper, we use the dropout mechanism (Srivastava et al., 2014) to emulate the ensemble process. Dropouts are typically used during the training phase only. In RCA, we employ the dropout mechanism during testing as well. Specifically, we use the networks of perturbed structures to perform multiple forward passes over the data in order to obtain a set of reconstruction losses for each test point. The final anomaly score is computed by averaging the reconstruction losses. Although dropout may increase the overall reconstruction loss, we expect a more robust estimation of the anomaly score using this procedure.

## 4 EXPERIMENTS

We have performed extensive experiments on both synthetic and real-world data to compare the performance of RCA against other baseline methods and to investigate its robustness to noise due to missing value imputation. The code is attached in the supplementary materials in submission.

### 4.1 RESULTS ON SYNTHETIC DATA

To better understand how RCA overcomes the limitations of conventional autoencoders (AE) on datasets with anomalies, we experimented with a synthetic 2-dimensional dataset. The dataset contains a pair of crescent-shaped moons with Gaussian noise (Pedregosa et al., 2011) representing the normal observations and anomalies generated from a 2-dimensional uniform distribution. In this experiment, we vary the proportion of anomalies from 10% to 40% while fixing the sample size to be 10,000. Samples with the top-$[(1 - \epsilon)n]$ highest anomaly scores are classified as anomalies, where $\epsilon$ is the anomaly ratio.

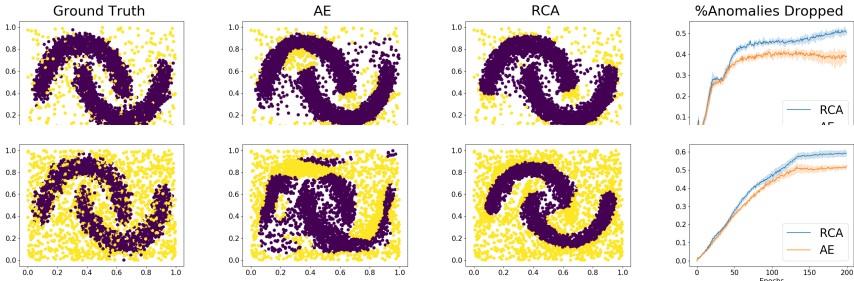

Figure 2: The first and second rows are results for 10% and 40% anomaly ratio, respectively. The last column shows the fraction of points with highest reconstruction loss that are true anomalies.

We show the results for 10% (top row) and 40% (bottom row) anomaly ratio[2] in Figure 2. Observe that the performance of both methods decreases with increasing anomaly ratio. However, results for RCA (third column) are more robust than AE (second column). In particular, when the anomaly ratio is 40%, AE fails to capture the true manifold of the normal data, unlike RCA. The result is consistent with Theorem 2, which states that, when anomaly ratio increases, using the subset of data selected by our algorithm is better than using all the data.

## 4.2 RESULTS ON REAL-WORLD DATA

For performance comparison, we use 18 benchmark datasets obtained from the Stony Brook ODDS library (Rayana, 2016). A summary description of the data is given in Table 2 in appendix. We reserve 60% of the data for training and the remaining 40% for testing. The performance of the competing methods are evaluated based on their Area under ROC curve (AUC) scores. We also performed experiments on the CIFAR10 dataset, for which the results are given in the Appendix.

We compared RCA against the following baseline methods: **SVDD** (deep one-class SVM) (Ruff et al., 2018), **VAE** (Variational autoencoder) (An & Cho, 2015; Kingma & Welling, 2013), **DAGMM** (deep gaussian mixture model) (Zong et al., 2018), **SO-GAAL** (Single-Objective Generative Adversarial Active Learning) (Liu et al., 2019), **OCSVM** (one-class SVM) (Chen et al., 2001), and **IF** (isolation forest) (Liu et al., 2008). Note that SVDD and DAGMM are two recent deep unsupervised AD methods while OCSVM and IF are two state-of-the-art AD methods. In addition, we also perform an ablation study to compare RCA against its four variants: **AE** (standard autoencoders without collaborative networks) and **RCA-E** (RCA without ensemble evaluation), and **RCA-SS** (RCA without sample selection). Since the methods are unsupervised, to ensure fair comparison, we maintain similar hyperparameter settings for all the competing DNN-based approaches to the best that we can (details can be found in the supplementary materials). Experimental results are reported based on their average AUC scores across 10 random initializations.

Figure 3a summarize the results of our experiments. The full table can be found in the Appendix. Note that RCA outperforms the deep unsupervised AD methods (SO-GAAL, DAGMM, SVDD) in 17 out of 18 datasets. These results suggest that the strategies employed by RCA are more effective at detecting anomalies compared to the ones used by the baseline deep unsupervised AD methods. Surprisingly, some of the more complex DNN baselines such as SO-GAAL, DAGMM, and SVDD perform poorly on the datasets. Their poor performance can be explained as follows. First, most of these baseline methods assume the availability of clean training data, whereas in our experiments, the training data was contaminated with anomalies to reflect more realistic settings. Second, we use the same network architecture on every datasets for all the methods (including RCA), since there is no guidance for tuning the network structure given that it is an unsupervised AD task. Finally, as will be discussed in Section 4.3, the performance of conventional unsupervised AD methods such as OCSVM and IF degrade significantly as the amount of missing values in the data increases, unlike the proposed RCA framework.

## 4.3 RESULTS FOR ABLATION STUDY AND ANOMALY DETECTION WITH MISSING VALUES

As real-world datasets are often imperfect, we compare the performance of RCA and other baseline methods in terms of their robustness to missing values. Mean imputation is a common approach to deal with missing values. In this experiment, we add missing values randomly in the features of

---

[2]More results for 20% and 30% can be found in the appendix

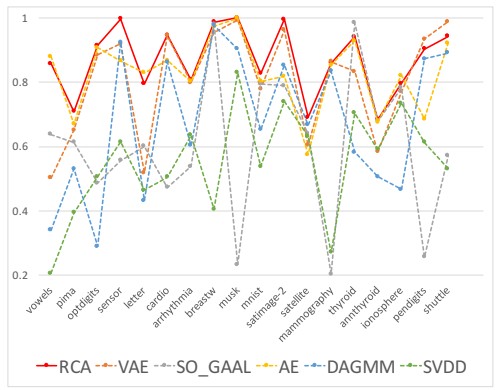

(a) AUC comparison of RCA against DNN base-lines. Details are given in Table 3 in Appendix.

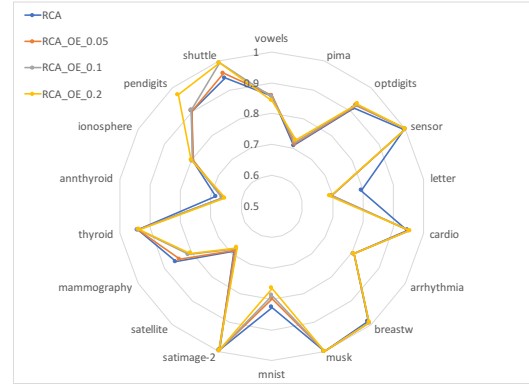

(b) Results of RCA for overestimating $\epsilon$ by 0.05, 0.1, 0.2. Details are given in Table 4 in Appendix.

Figure 3: Experimental results on 18 benchmark datasets from ODDS repository.

| Missing Ratio | RCA-E | RCA-SS | VAE | SO-GAAL | AE | DAGMM | SVDD | OCSVM | IF |
|---|---|---|---|---|---|---|---|---|---|
| 0.0 | 11-2-5 | 16-0-2 | 14-0-4 | 17-0-1 | 15-0-3 | 18-0-0 | 18-0-0 | 10-1-7 | 10-0-8 |
| 0.1 | 12-1-5 | 14-1-3 | 16-1-1 | 16-0-2 | 14-0-4 | 17-0-1 | 18-0-0 | 13-1-4 | 12-0-6 |
| 0.2 | 11-1-6 | 13-3-2 | 14-2-2 | 17-0-1 | 13-0-5 | 18-0-0 | 18-0-0 | 15-0-3 | 9-0-9 |
| 0.3 | 9-3-6 | 13-1-4 | 15-0-3 | 17-1-0 | 13-0-5 | 18-0-0 | 18-0-0 | 16-0-2 | 14-1-3 |
| 0.4 | 10-0-8 | 12-2-4 | 14-0-4 | 15-0-3 | 12-0-6 | 17-0-1 | 18-0-0 | 16-0-2 | 15-0-3 |
| 0.5 | 8-3-7 | 10-1-7 | 11-1-6 | 14-0-4 | 9-0-9 | 15-0-3 | 17-0-1 | 14-1-3 | 13-0-5 |

Table 1: Comparison of RCA against various competing methods in terms of (#win-#draw-#loss) on 18 benchmark datasets with different missing ratios. RCA-E (no ensemble), RCA-SS (no sample selection), AE (no ensemble and no sample selection) are used for ablation study of our method.

each benchmark dataset and apply mean imputation to replace the missing values. Such imputation process will likely introduce noise into the data. We vary the percentage of missing values from 10% to 50% and compare the average AUC scores of the competing methods. The results are summarized in Table 1, which shows the number of wins, draws, and losses of RCA compared to each baseline method on the 18 benchmark datasets. We also include results from ablation study to investigate the effectiveness of using sample selection and ensemble evaluation. Specifically, we compare RCA against its variants, RCA-E, RCA-SS, and AE. The results show that our framework is better than the baselines on the majority of the datasets in almost all settings. In particular, RCA consistently outperforms both DAGMM and SVDD by more than 80%, demonstrating the robustness of our algorithm compared to other deep unsupervised AD methods when training data is contaminated. Additionally, as the missing ratio increases to more than 30%, it outperforms IF and OCSVM by more than 70% on the datasets. On the other hand, the advantage of RCA over its variants, AE, RCA-SS, and RCA-E, is significant when the missing ratio is less than 40%, but becomes less significant at higher missing ratios. Finally, since the true anomaly ratio $\epsilon$ is often unknown in practice, we conducted experiments to evaluate the robustness of RCA when $\epsilon$ is overestimated by 5%, 10%, or 20% from their true values on all the datasets. Fig. 3b shows the AUC scores for RCA do not change significantly even when $\epsilon$ was highly overestimated on most of the datasets.

## 5 CONCLUSION

This paper introduces RCA, a robust collaborative autoencoder framework for unsupervised AD. The framework is designed to overcome limitations of existing deep unsupervised AD methods due to over-parameterization of the DNNs, which hampers their effectiveness. We theoretically show the effectiveness of our algorithm to prevent model overfitting due to anomalies. In addition, we empirically show that RCA outperforms various state-of-the-art unsupervised AD algorithms in most experimental settings. We also found RCA to be more robust to noise introduced by missing value imputation compared to other baseline methods. In the future, we aim to extend the proposed framework to incorporate more than two autoencoders. We will also investigate whether it is possible to relax some of the assumptions behind our theoretical bounds to more realistic scenarios.

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

## A  APPENDIX

### A.1  DATA STATISTICS

The data statistics are in table 2.

### A.2  FULL TABLE RESULTS ON ODDS

See table 3.

### A.3  DISCUSSION ABOUT SVDD AND DAGMM

The reason why SVDD and DAGMM performs bad in ODDS might have several reasons. First, the results reported in the SVDD paper assume training data has no contamination. For the DAGMM paper, it has only 2 datasets that overlaps with our experiments (thyroid, arrhythmia), for which the results reported are also for clean data. In contrast, the training data used in our experiments are contaminated with anomalies. Also, the DAGMM paper acknowledges that their performance degrades when the amount of contamination increases. Furthermore, unlike our experiments, the results reported in the DAGMM paper use different network structure for each dataset to obtain good performance, this is because that they have clean dataset, which makes it possible to tuning the network structure and hyperparameters. However, choosing the right network structure is

impractical for unsupervised anomaly detection without ground truth labels available. In our experiments, all baselines and our method use the same network structure across different datasets. Since there is no official code of DAGMM from the authors, our implementation of DAGMM are highly depends on these two open source implementations [3]. Another question people may ask is that why complexed deep methods such as DAGMM, SVDD, SO-GAAL cannot beat shallow methods such as OCSVM and Isolation Forest. According to our best knowledge, there is no evidence SVDD, SO-GAAL and DAGMM performed better than OCSVM and IF on datasets beyond the benchmark image data (CIFAR-10). In fact, the SO-GAAL, OCSVM and IF results reported in our paper for ODDS dataset are consistent with the numbers reported in PyOD [4], a Python toolkit for outlier detection. Also, Reference [8] compared DAGMM, SVDD (denoted as E2E), against OCSVM. The results shown in Table 1 of the paper are similar to ours. In order to make sure that our algorithm is better than SVDD or there is nothing wrong in our implementation about SVDD, we also conduct experiments on CIFAR10. Since our methods are not specifically designed for the image data, we process the CIFAR10 results in following way:

We use pytorch official implmentation of vgg19 (pretrained on ImageNet) to extract 4096 dimensional feature representation of CIFAR10 to perform the anomaly detection for each class (10 sub-datasets). Each sub-datasets consists 5000 normal class and 5% anomalies, which are random sampled from other class. The training, testing data consists 80% and 20% of data (i.e. training data has (5000 + 250) * 0.8 = 4200 samples, testing data has 1050 samples). All results are averaged over 5 random seeds. The results are in the figure 4. For RCA, DAGMM, SVDD, we use the same network structure in our paper. For SVDD_Original, we directly borrow the number from original SVDD paper (Ruff et al., 2018).

### A.4 PROOF OF THEOREM 1

denote $M = n(1 - \epsilon)L_{max}$ to be the smoothness of function $F = \sum_{i \notin \mathcal{O}} f_i(\mathbf{w})$, denote the update rule $\mathbf{w}^{(t+1)} = \mathbf{w}^{(t)} - \eta^{(t)} \sum_i p_i^{(t)} \nabla f_i(\mathbf{w})$. For the normal stochastic gradient descent, by smoothness, we have:

$$F(\mathbf{w}^{(t+1)}) - F(\mathbf{w}^{(t)}) \leq \langle \nabla F(\mathbf{w}^{(t)}), \mathbf{w}^{(t+1)} - \mathbf{w}^{(t)} \rangle + \frac{M}{2} \| \mathbf{w}^{(t)} - \mathbf{w}^{(t+1)} \|^2$$

$$\leq -\eta^{(t)} \langle \nabla F(\mathbf{w}^{(t)}), \nabla f_i(\mathbf{w}^{(t)}) \rangle + \frac{\eta^{(t)^2} M}{2} \| \nabla f_i(\mathbf{w}^{(t)}) \|_2^2 \qquad (4)$$

---

[3] https://github.com/danieltan07/dagmm, https://github.com/tnakae/DAGMM

[4] We use the pyod implmentation (https://github.com/yzhao062/pyod) of SO-GAAL, VAE, IF, OCSVM

Table 2: Summary of benchmark data, where $N$ is sample size and $d$ is number of features.

| Dataset | $N$ | $d$ | Anomaly ratio |
|---|---|---|---|
| vowels | 1456 | 12 | 3.4% |
| pima | 768 | 8 | 35% |
| optdigits | 5216 | 64 | 3% |
| sensor | 58509 | 48 | 9.1% |
| letter | 1600 | 32 | 6.25% |
| cardio | 1831 | 21 | 9.6% |
| arrhythmia | 452 | 274 | 15% |
| breastw | 683 | 9 | 35% |
| musk | 3062 | 166 | 3.2% |
| mnist | 7603 | 100 | 9.2% |
| satimage-2 | 5803 | 36 | 1.2% |
| satellite | 6435 | 36 | 32% |
| mammography | 11183 | 6 | 2.32% |
| thyroid | 3772 | 6 | 2.5% |
| annthyroid | 7200 | 6 | 7.42% |
| ionosphere | 351 | 33 | 36% |
| pendigits | 6870 | 16 | 2.27% |
| shuttle | 49097 | 9 | 7% |

Table 3: Performance comparison of RCA against various baseline methods in terms of average AUC scores and its standard deviation across 10 random seeds.

| Dataset | RCA | VAE | SO-GAAL | AE | DAGMM | SVDD | OCSVM | IF |
|---|---|---|---|---|---|---|---|---|
| vowels | 0.857±0.03 | 0.50±0.043 | 0.637±0.20 | **0.879±0.02** | 0.340±0.10 | 0.206±0.04 | 0.765±0.04 | 0.768±0.01 |
| pima | **0.709±0.01** | 0.651±0.02 | 0.613±0.05 | 0.669±0.01 | 0.531±0.03 | 0.395±0.03 | 0.594±0.03 | 0.662±0.02 |
| optdigits | **0.914±0.02** | 0.768±0.01 | 0.487±0.14 | 0.907±0.01 | 0.290±0.04 | 0.506±0.02 | 0.558±0.01 | 0.706±0.04 |
| sensor | **0.996±0.01** | 0.918±0.00 | 0.557±0.22 | 0.866±0.05 | 0.924±0.08 | 0.614±0.07 | 0.939±0.00 | 0.948±0.00 |
| letter | 0.795±0.05 | 0.517±0.04 | 0.601±0.06 | **0.829±0.03** | 0.433±0.03 | 0.465±0.04 | 0.557±0.04 | 0.643±0.04 |
| cardio | **0.946±0.01** | 0.945±0.01 | 0.473±0.08 | 0.867±0.02 | 0.862±0.03 | 0.505±0.06 | 0.936±0.00 | 0.927±0.01 |
| arrhythmia | **0.807±0.05** | 0.798±0.04 | 0.538±0.04 | 0.802±0.04 | 0.603±0.09 | 0.635±0.06 | 0.782±0.03 | 0.802±0.02 |
| breastw | **0.986±0.01** | 0.953±0.01 | 0.980±0.01 | 0.973±0.00 | 0.976±0.00 | 0.406±0.04 | 0.955±0.01 | 0.983±0.01 |
| musk | **1.000±0.00** | 0.764±0.01 | 0.234±0.19 | 0.998±0.00 | 0.903±0.13 | 0.829±0.05 | **1.000±0.00** | 0.995±0.01 |
| mnist | 0.827±0.02 | **0.847±0.00** | 0.795±0.02 | 0.802±0.01 | 0.652±0.08 | 0.538±0.05 | 0.835±0.01 | 0.800±0.01 |
| satimage-2 | 0.995±0.00 | 0.962±0.01 | 0.789±0.18 | 0.818±0.07 | 0.853±0.11 | 0.739±0.09 | **0.998±0.00** | 0.996±0.00 |
| satellite | 0.69±0.01 | 0.603±0.01 | 0.640±0.07 | 0.575±0.07 | 0.667±0.19 | 0.631±0.02 | 0.650±0.01 | **0.700±0.03** |
| mammography | 0.859±0.01 | 0.863±0.01 | 0.204±0.03 | 0.853±0.02 | 0.834±0.00 | 0.272±0.01 | **0.881±0.02** | 0.873±0.02 |
| thyroid | 0.941±0.01 | 0.836±0.01 | **0.984±0.01** | 0.928±0.02 | 0.582±0.09 | 0.704±0.03 | 0.960±0.01 | 0.980±0.01 |
| annthyroid | 0.684±0.02 | 0.583±0.02 | 0.679±0.02 | 0.675±0.02 | 0.506±0.02 | 0.591±0.01 | 0.599±0.01 | **0.824±0.01** |
| ionosphere | 0.796±0.01 | 0.760±0.01 | 0.783±0.08 | 0.821±0.01 | 0.467±0.08 | 0.735±0.05 | 0.812±0.04 | **0.843±0.02** |
| pendigits | 0.903±0.02 | 0.931±0.00 | 0.257±0.05 | 0.685±0.07 | 0.872±0.07 | 0.613±0.07 | 0.935±0.00 | **0.941±0.01** |
| shuttle | 0.942±0.01 | 0.988±0.00 | 0.571±0.32 | 0.921±0.01 | 0.890±0.11 | 0.531±0.29 | 0.985±0.00 | **0.997±0.00** |
| glass | 0.625±0.13 | 0.595±0.14 | 0.420±0.11 | 0.570±0.15 | **0.852±0.08** | 0.756±0.11 | 0.522±0.21 | 0.706±0.06 |

Table 4: Sensitivity Analysis about $\epsilon$, the first number is averaged auc score and the second number is the standard deviation. All experiments are repeated for 10 random seeds.

| Dataset | $\epsilon$ | $\epsilon + 0.05$ | $\epsilon + 0.1$ | $\epsilon + 0.2$ |
|---|---|---|---|---|
| vowels | 0.857±0.03 | 0.858±0.03 | 0.855±0.03 | 0.842±0.03 |
| pima | 0.709±0.01 | 0.715±0.01 | 0.721±0.01 | 0.728±0.01 |
| optdigits | 0.914±0.02 | 0.926±0.01 | 0.927±0.01 | 0.932±0.01 |
| sensor | 0.996±0.01 | 1.000±0.00 | 1.000±0.00 | 1.000±0.00 |
| letter | 0.795±0.05 | 0.698±0.04 | 0.697±0.04 | 0.691±0.05 |
| cardio | 0.946±0.01 | 0.951±0.01 | 0.953±0.01 | 0.954±0.00 |
| arrhythmia | 0.807±0.05 | 0.807±0.05 | 0.807±0.05 | 0.807±0.05 |
| breastw | 0.986±0.01 | 0.992±0.00 | 0.992±0.00 | 0.991±0.00 |
| musk | 1.000±0.00 | 1.000±0.00 | 1.000±0.00 | 1.000±0.00 |
| mnist | 0.827±0.02 | 0.799±0.02 | 0.789±0.02 | 0.765±0.01 |
| satimage-2 | 0.995±0.00 | 0.998±0.00 | 0.998±0.00 | 0.998±0.00 |
| satellite | 0.69±0.01 | 0.687±0.00 | 0.678±0.01 | 0.675±0.03 |
| mammography | 0.859±0.01 | 0.844±0.01 | 0.811±0.01 | 0.804±0.01 |
| thyroid | 0.941±0.01 | 0.939±0.01 | 0.938±0.01 | 0.935±0.01 |
| annthyroid | 0.684±0.02 | 0.661±0.01 | 0.663±0.02 | 0.654±0.01 |
| ionosphere | 0.796±0.01 | 0.794±0.01 | 0.796±0.01 | 0.801±0.01 |
| pendigits | 0.903±0.02 | 0.903±0.03 | 0.905±0.03 | 0.971±0.01 |
| shuttle | 0.942±0.01 | 0.958±0.03 | 0.994±0.00 | 0.994±0.00 |

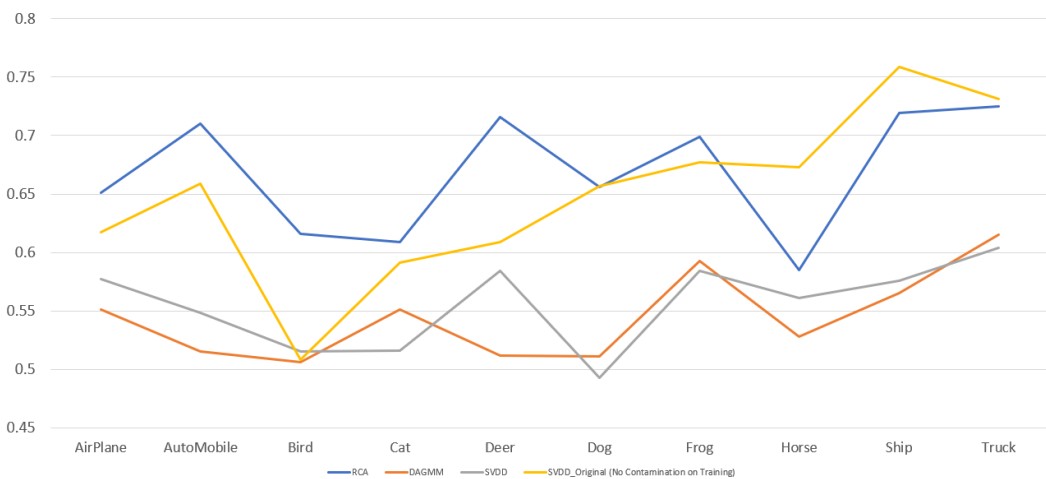

Figure 4: Results on CIFAR10 for SVDD, DAGMM, and our method. x-axis are the normal class while y-axis is the auc score. SVDD are our implementation on contaminated data described above. The SVDD-Original is the number reported in SVDD original paper (where they use clean training data). We could see that even use contaminated data, our model still outperforms deep svdd in most settings, and SVDD performs bad when training data has contamination. (i.e. SVDD performs much worse than SVDD_Original)

Take expectation on $\nabla f_i(\mathbf{w}^{(t)})$ by our sampling probability and applying triangle inequality on the last term with inequality $\sum p_i^2 \leq \sum p_i$, we have

$$\mathbb{E}F(\mathbf{w}^{t+1}) - F(\mathbf{w}^{(t)}) \leq -\eta^{(t)}\langle\nabla F(\mathbf{w}^{(t)}), \sum_i p_i(\mathbf{w}^{(t)})\nabla f_i(\mathbf{w}^{(t)})\rangle + \sum_i p_i(\mathbf{w}^{(t)})\frac{\eta^{(t)^2}M}{2}\|\nabla f_i(\mathbf{w}^{(t)})\|^2$$

$$\leq -\eta^{(t)}\langle\nabla F(\mathbf{w}^{(t)}), \sum_i p_i(\mathbf{w}^{(t)})\nabla f_i(\mathbf{w}^{(t)}) - \nabla F(\mathbf{w})\rangle - \eta^{(t)}\|\nabla F(\mathbf{w})\|^2 + \eta^{(t)^2}\frac{MG^2}{2}$$

complete the square and let $\hat{F}(\mathbf{w}^{(t)}) = \sum_i p_i(\mathbf{w}^{(t)})\nabla f_i(\mathbf{w}^{(t)})$, we have

$$\leq \frac{\eta^{(t)}}{2}\|\nabla F(\mathbf{w}^{(t)})\|^2 + \frac{\eta^{(t)}}{2}\|\nabla F(\mathbf{w}^{(t)}) - \nabla\hat{F}(\mathbf{w}^{(t)})\|^2 - \eta^{(t)}\|\nabla F(\mathbf{w}^{(t)})\|^2 + \frac{\eta^{(t)^2}MG^2}{2}$$

$$\leq \frac{\eta^{(t)}}{2}\|\nabla F(\mathbf{w}^{(t)}) - \nabla\hat{F}(\mathbf{w}^{(t)})\|^2 - \frac{\eta^{(t)}}{2}\|\nabla F(\mathbf{w}^{(t)})\|^2 + \frac{\eta^{(t)^2}MG^2}{2}$$

Move the gradient norm to the left, and take total expectation, we have

$$\eta^{(t)}\mathbb{E}\|\nabla F(\mathbf{w}^{(t)})\|^2 \leq 2\left(\mathbb{E}F(\mathbf{w}^{(t)}) - \mathbb{E}F(\mathbf{w}^{(t+1)})\right) + \eta^{(t)}\mathbb{E}\|\nabla F(\mathbf{w}^{(t)}) - \nabla\hat{F}(\mathbf{w}^{(t)})\|^2 + \eta^{(t)^2}MG^2$$

Sum it from $t = 0$ to $t = T$, we have:

$$\sum_{t=0}^{T}\eta^{(t)}\mathbb{E}\|\nabla F(\mathbf{w}^{(t)})\|^2 \leq 2\left(\mathbb{E}F(\mathbf{w}^{(0)}) - \mathbb{E}F(\mathbf{w}^{(T+1)})\right) + \sum_{t=0}^{T}\eta^{(t)}\mathbb{E}\|\nabla F(\mathbf{w}^{(t)}) - \nabla\hat{F}(\mathbf{w}^{(t)})\|^2 + \eta^{(t)^2}MG^2$$

$$\leq 2B + \sum_{t=0}^{T}\eta^{(t)}C + \sum_{t=0}^{T}\eta^{(t)^2}MG^2$$

$$\min_{t=0,1,2,...,T}\mathbb{E}\|\nabla F(\mathbf{w}^{(t)})\|^2 \leq \mathbb{E}\|\nabla F(\mathbf{w}^{(t)})\|^2 \leq \frac{2B}{\sum\eta^{(t)}} + MG^2\frac{\sum\eta^{(t)^2}}{\sum\eta^{(t)}} + C \quad (5)$$

By using the assumption of learning rate ($\sum\eta^{(t)} = \infty, \sum\eta^{(t)^2} \leq \infty$), the first two term can be ignored when $T$ goes to infinity. We can get the convergence in theorem 1 (The convergence

rate $\log(T)$ is get by assume learning rate is $\eta^{(t)} = 1/t$, which satisfy the above learning rate assumption).

We also provided a better convergence rate compared to submitted manuscript with a stricter learning rate setting. Start from equation 4, we have:

$$\mathbb{E}F(\mathbf{w}^{t+1}) - F(\mathbf{w}^{(t)}) \leq -\eta^{(t)}\langle \nabla F(\mathbf{w}^{(t)}), \sum_i p_i(\mathbf{w}^{(t)})\nabla f_i(\mathbf{w}^{(t)})\rangle + \sum_i p_i(\mathbf{w}^{(t)})\frac{\eta^{(t)^2}M}{2}\|\nabla f_i(\mathbf{w}^{(t)})\|^2$$

$$\leq -\eta^{(t)}\langle \nabla F(\mathbf{w}^{(t)}), \sum_i p_i(\mathbf{w}^{(t)})\nabla f_i(\mathbf{w}^{(t)}) - \nabla F(\mathbf{w})\rangle - \eta^{(t)}\|\nabla F(\mathbf{w})\|^2 + \eta^{(t)^2}\frac{MG^2}{2}$$

complete square in a different way and let $\hat{F}(\mathbf{w}^{(t)}) = \sum_i p_i(\mathbf{w}^{(t)})\nabla f_i(\mathbf{w}^{(t)})$, we have

$$\leq \eta^{(t)}\frac{1}{2}\left(\eta^{(t)}\|\nabla F(\mathbf{w}^{(t)})\|^2 + \frac{1}{\eta^{(t)}}\|\nabla F(\mathbf{w}^{(t)}) - \nabla\hat{F}(\mathbf{w}^{(t)})\|^2\right)$$

$$- \eta^{(t)}\|\nabla F(\mathbf{w}^{(t)})\|^2 + \frac{\eta^{(t)^2}MG^2}{2}$$

$$\leq \frac{\eta^{(t)^2}}{2}\|\nabla F(\mathbf{w}^{(t)})\|^2 + \frac{1}{2}\|\nabla F(\mathbf{w}^{(t)}) - \nabla\hat{F}(\mathbf{w}^{(t)})\|^2 - \eta^{(t)}\|\nabla F(\mathbf{w}^{(t)})\|^2 + \frac{\eta^{(t)^2}MG^2}{2}$$

$$\leq \frac{1}{2}\|\nabla F(\mathbf{w}^{(t)}) - \nabla\hat{F}(\mathbf{w}^{(t)})\|^2 - \eta^{(t)}\|\nabla F(\mathbf{w}^{(t)})\|^2 + \frac{\eta^{(t)^2}(M+1)G^2}{2}$$

Move the gradient norm to the left, and take total expectation, we have

$$\eta^{(t)}\mathbb{E}\|\nabla F(\mathbf{w}^{(t)})\|^2 \leq \mathbb{E}F(\mathbf{w}^{(t)}) - \mathbb{E}F(\mathbf{w}^{(t+1)}) + \frac{1}{2}\mathbb{E}\|\nabla F(\mathbf{w}^{(t)}) - \nabla\hat{F}(\mathbf{w}^{(t)})\|^2 + \frac{\eta^{(t)^2}(M+1)G^2}{2}$$

Sum it from $t = 0$ to $t = T$, we have:

$$\sum_{t=0}^{T}\eta^{(t)}\mathbb{E}\|\nabla F(\mathbf{w}^{(t)})\|^2 \leq \mathbb{E}F(\mathbf{w}^{(0)}) - \mathbb{E}F(\mathbf{w}^{(T+1)})$$

$$+ \sum_{t=0}^{T}\frac{1}{2}\mathbb{E}\|\nabla F(\mathbf{w}^{(t)}) - \nabla\hat{F}(\mathbf{w}^{(t)})\|^2 + \sum_{t=0}^{T}\frac{\eta_t^2(M+1)G^2}{2}$$

$$\leq B + \frac{TC}{2} + \sum_{t=0}^{T}\frac{\eta^{(t)^2}(M+1)G^2}{2}$$

$$\min_{t=0,1,2,\ldots,T}\mathbb{E}\|\nabla F(\mathbf{w}^{(t)})\|^2 \leq \frac{B}{\sum\eta^{(t)}} + \frac{(M+1)G^2}{2}\frac{\sum\eta^{(t)^2}}{\sum\eta^{(t)}} + \frac{TC}{\sum\eta^{(t)}}$$

By assume the learning rate is constant, we write the RHS as a function of learning rate

$$f(\eta) = \frac{B}{T\eta} + \frac{(M+1)G^2\eta}{2} + \frac{TC}{T\eta}$$

$$= \frac{B}{T\eta} + \frac{(M+1)G^2\eta}{2} + \frac{C}{\eta}$$

Let $(M+1)G^2 = H$, we have

$$= \frac{B}{T\eta} + \frac{H\eta}{2} + \frac{C}{\eta}$$

We study the minima of function:

$$f(x) = \frac{a}{x} + bx$$

$$x^* = \sqrt{(a/b)}, f(x^*) = 2(\sqrt{ab})$$

Thus, to minimize the bound in RHS, we have the optimal learning rate $\eta^* = \sqrt{\dfrac{2B}{HT} + \dfrac{2C}{H}}$ by

letting $a = \dfrac{B}{T} + C, b = \dfrac{H}{2}$. Then, the optimal value of RHS is:

$$f(\eta^*) = 2(\sqrt{(\frac{B}{T} + C)(\frac{H}{2})}) = \sqrt{\frac{2BH}{T} + 2CH} \leq \sqrt{\frac{2BH}{T}} + \sqrt{2CH} = \mathbb{O}(\frac{1}{\sqrt{T}}) + \mathbb{O}(\sqrt{C})$$

. Thus we can conclude that $\min_{t=0,1,2,\ldots,T} \mathbb{E}\|\nabla F(\mathbf{w}^{(t)})\|^2 \to \mathbb{O}(\dfrac{1}{\sqrt{T}}) + \mathbb{O}(\sqrt{C})$.

This is even a better results compared to theorem 1 in our paper, since we could achieve the convergence rate of $\sqrt{\dfrac{1}{T}}$ compared to $\log(T)$, while the error term is dependent on $\sqrt{C}$ instead of $C$. However, to achieve this rate, we need stricter condition on learning rate.

### A.5   PROOF OF THEOREM 2

We analysis the stationary point of using all data. Let $\mathbf{w}^*_{ns}$ denotes the stationary point by using the entire data, $\mathbf{w}^*$ denotes the stationary point by using the clean data, by the stationary condition of $\mathbf{w}^*_{ns}$, we have

$$\sum_{i\notin\mathcal{O}}^{p} \nabla f_i(\mathbf{w}^*_{ns}) = -\sum_{j\in\mathcal{O}}^{q} \nabla f_j(\mathbf{w}^*_{ns})$$

$$\|\sum_{i\notin\mathcal{O}}^{p} \nabla f_i(\mathbf{w}^*_{ns})\| = \|\sum_{j\in\mathcal{O}}^{q} \nabla f_j(\mathbf{w}^*_{ns})\|$$

Upper bound for LHS

$$\|\sum_{i\notin\mathcal{O}}^{p} \nabla f_i(\mathbf{w}^*_{ns})\| = \|\sum_{i\notin\mathcal{O}}^{p} \nabla f_i(\mathbf{w}^*_{ns}) - \sum_{i\notin\mathcal{O}}^{p} \nabla f_i(\mathbf{w}^*)\| \leq \sum_{i\notin\mathcal{O}}^{p} \|\nabla f_i(\mathbf{w}^*_{ns}) - \nabla f_i(\mathbf{w}^*)\| \leq \sum_{i\notin\mathcal{O}}^{p} L_i\|\mathbf{w}^*_{nsi} - \mathbf{w}^*\|$$

$$= (1-\epsilon)nL_{max} \max_i \|\mathbf{w}^*_{nsi} - \mathbf{w}^*\|$$

$\leq M\delta$   ($\delta$ is defined in equation 3 in the submitted manuscript)

Another upper bound for LHS from RHS

$$\|\sum_{i\notin\mathcal{O}}^{p} \nabla f_i(\mathbf{w}^*_{ns})\| = \|\sum_{j\in\mathcal{O}}^{q} \nabla f_j(\mathbf{w}^*_{ns})\| \leq n\epsilon G$$

Thus, we have

$$\|\nabla F(\mathbf{w}^*_{ns})\| \leq \min(n\epsilon G, M\delta) \tag{6}$$

From theorem 1, by setting $\eta^* = \sqrt{\dfrac{2B}{HT} + \dfrac{2C}{H}}$, it is trivial to get

$$\min_{t=0,1,2,\ldots,T} \mathbb{E}\|\nabla F(\mathbf{w}^{(t)})\|^2 \leq \sqrt{\frac{2BH}{T} + 2CH}$$

Or by assumption of $\sum \eta^{(t)} = \infty, \sum \eta^{(t)} \leq \infty$, from equation 5 we can get

$$\min_{t=0,1,2,\ldots,T} \mathbb{E}\|\nabla F(\mathbf{w}^{(t)})\|^2 \leq \mathbb{E}\|\nabla F(\mathbf{w}^{(t)})\|^2 \leq \mathbb{O}(\frac{1}{\sum \eta^{(t)}}) + \mathbb{O}(\frac{\sum \eta^{(t)^2}}{\sum \eta^{(t)}}) + C$$

We would like to study the worst case, when the upper bound of our algorithm is better than the upper bound without sample selection.

Thus, we want the following holds when $t$ goes to infinity

$$\sqrt{\frac{2BH}{T} + 2CH} \leq \min(n\epsilon G, M\delta)^2$$

When $t$ goes to infinity, by the assumption we have for learning rate, we have:

$$\sqrt{C} \leq \sqrt{1/2H}(\min(n\epsilon G, M\delta))^2 \text{(fixed optimal lr)}$$

Or similarly, in deminishing learning rate setting, we have

$$C \leq (\min(n\epsilon G, M\delta))^2 (\sum \eta^{(t)} = \infty, \sum \eta^{(t)^2} \leq \infty)$$

Thus both LHS and RHS are the upper bound, thus we could get the conclusion that in worst cases, our solution is better than algorithm without sample selection, which gets our conclusion of existence.

### A.6   PROOF OF THEOREM 3

Now, we try to prove the correctness of the algorithm. We assume $\mathbf{w}^*$ satisfy $f_i(\mathbf{w}^*) < f_j(\mathbf{w}^*), \forall \mathbf{x}_i \notin \mathcal{O}, \mathbf{x}_j \in \mathcal{O}$.

Without loss of generality, we could define $\delta, \phi \geq 1$ as below:

$$\delta \leq \|\mathbf{w}_j^* - \mathbf{w}^*\| \leq \phi\delta, \forall j \in \mathcal{O} \tag{7}$$

Now, we try to answer the first question, under what conditions, our solution is perfect. We define some neighbors around the optimal point. According to our anomaly detection setting, we with those anomalies should have higher loss compared to the normal data point:

$$\mathcal{B}_r\left(\boldsymbol{w}^*\right) = \left\{ \begin{array}{l} \boldsymbol{w} | f_i(\boldsymbol{w}) < f_j(\boldsymbol{w}), \forall i \notin \mathbb{O}, j \in \mathbb{O}, \\ \|\boldsymbol{w} - \boldsymbol{w}^*\| \leq r \end{array} \right\}. \tag{8}$$

We knew that such ball with radius $r$ must be existed since the loss function above is both strongly smooth.

In order to better describe $\mathcal{B}_r\left(\boldsymbol{w}^*\right)$, we would like to analysis the boundary of the ball.

Denote the set of intersection between the loss surface of the normal data and the loss surface of the abnormal data as:

$$\Omega_{\mathbf{w}} = \left\{\mathbf{w}_{ij} \big| f_{\mathbf{x}_i \notin \mathcal{O}}(\mathbf{w}) = f_{\mathbf{x}_j \in \mathcal{O}}(\mathbf{w})\right\} \tag{9}$$

Then, we can write the boundary point of the ball as:

$$\mathbf{w}_B = \arg\min_{w \in \Omega_{\mathbf{w}}} \|\mathbf{w}_B - \mathbf{w}^*\| \tag{10}$$

At $\mathbf{w}_B$, by using smoothness and convexity, we have

$$f_i(\mathbf{w}_B) \leq f_i(\mathbf{w}^*) + \langle \mathbf{w}_B - \mathbf{w}^*, \nabla f_i(\mathbf{w}^*)\rangle + \frac{L_i}{2}\|\mathbf{w}_B - \mathbf{w}^*\|^2$$

$$f_j(\mathbf{w}_B) \geq f_j(\mathbf{w}_j^*) + \langle \mathbf{w}_B - \mathbf{w}_j^*, \nabla f_j(\mathbf{w}^*)\rangle + \frac{\mu_j}{2}\|\mathbf{w}_B - \mathbf{w}_j^*\|^2$$

By the definition of $\mathbf{w}_B$ and equal minimum assumption, without loss of generality, we could assume the minimum is 0, which does not affect the results. Then, we have

$$f_i(\mathbf{w}_B) \leq \frac{L_i}{2}\|\mathbf{w}_B - \mathbf{w}^*\|^2$$

$$-f_j(\mathbf{w}_B) \leq -\frac{\mu_j}{2}\|\mathbf{w}_B - \mathbf{w}_j^*\|^2$$

Adding two inequality, we have:

$$\|\mathbf{w}_B - \mathbf{w}^*\|^2 \geq \frac{\mu_j}{L_i}\|\mathbf{w}_B - \mathbf{w}_j^*\|^2$$

By triangle inequality, we have

$$\|\mathbf{w}_B - \mathbf{w}^*\| + \|\mathbf{w}_B - \mathbf{w}_j^*\| \geq \|\mathbf{w}_j^* - \mathbf{w}^*\|$$

Combining above two inequalities, we have

$$\|\mathbf{w}_B - \mathbf{w}^*\| \geq \sqrt{\frac{\mu_j}{L_i}}(\|\mathbf{w}_j^* - \mathbf{w}^*\| - \|\mathbf{w}_B - \mathbf{w}^*\|)$$

$$(1 + \sqrt{\frac{\mu_j}{L_i}})\|\mathbf{w}_B - \mathbf{w}^*\| \geq \sqrt{\frac{\mu_j}{L_i}}\delta$$

$$\text{let } \kappa = \sqrt{\frac{L_{max}^c}{\mu_{min}^o}}$$

$$\|\mathbf{w}_B - \mathbf{w}^*\| \geq \frac{1}{1 + \sqrt{\frac{L_i}{\mu_j}}}\delta \geq \frac{1}{1 + \kappa}\delta$$

, where $L_{max}^c$ denotes the maximum lipschitz smoothness in clean data and $\mu_{min}^o$ denotes the minimum convexity in anomalies.

Define $F(\mathbf{w}) = \sum_{\mathbf{x}_i \notin O} f_i(\mathbf{w})$, we have that function $F(\mathbf{w})$ satisfy $m = n(1 - \epsilon)\mu_{min}$ convexity. Similarly, we know that $F(\mathbf{w})$ also satisfy $M = n(1 - \epsilon)L_{max}$ smoothness. Then, we can have:

$$\|\mathbf{w}_{sr}^* - \mathbf{w}^*\| \leq m\|\nabla F(\mathbf{w}_{sr}^*) - \nabla F(\mathbf{w}^*)\|$$
$$= m\|\nabla F(\mathbf{w}_{sr}^*)\|$$

$$\|\mathbf{w}_{sr}^* - \mathbf{w}^*\|^2 \leq m^2\|\nabla F(\mathbf{w}_{sr}^*)\|^2$$

Now, our goal is trying to upper bound the term $\|\nabla F(\mathbf{w}_{sr}^*)\|$. According to theorem 1 for fixed optimal learning rate, we have

$$\min_{t=0,1,2,...,T} \mathbb{E}\|\nabla F(\mathbf{w}^{(t)})\|^2 \leq \mathbb{E}\|\nabla F(\mathbf{w}^{(t)})\|^2 \leq \sqrt{\frac{2BH}{T} + 2CH}$$

Now, we have:

$$\|\mathbf{w}_{sr}^* - \mathbf{w}^*\|^2 \leq m^2\sqrt{\frac{2BH}{T} + 2CH}$$

$$\|\mathbf{w}_B - \mathbf{w}^*\|^2 \geq (\frac{1}{1 + \kappa}\delta)^2$$

Thus, we could get the sufficient condition for $\|\mathbf{w}_{sr}^* - \mathbf{w}^*\|^2 \leq \|\mathbf{w}_B - \mathbf{w}^*\|^2$ as

$$m^2\sqrt{\frac{2BH}{T} + 2CH} \leq (\frac{1}{1 + \kappa}\delta)^2$$

by using the optimal fixed learning rate and assume $T$ is sufficiently large, rearrange the term, we have:

$$\sqrt{C} \leq \frac{1}{\sqrt{2H}}(\frac{\delta}{(1 + \kappa)m})^2 = \mathbb{O}(\frac{\delta}{\kappa})^2$$

$$C \leq \mathbb{O}(\frac{\delta}{\kappa})^4$$

Similarly, from the theorem 1 for the diminishing learning rate condition $\sum \eta^{(t)} = \infty, \sum \eta^{(t)} \leq \infty$, we have

$$C \leq (\frac{\delta}{(1 + \kappa)m})^2 = \mathbb{O}(\frac{\delta}{\kappa})^2$$

We can conclude that as long as the above inequality holds, we can guarantee that our algorithm returns the correct answer.

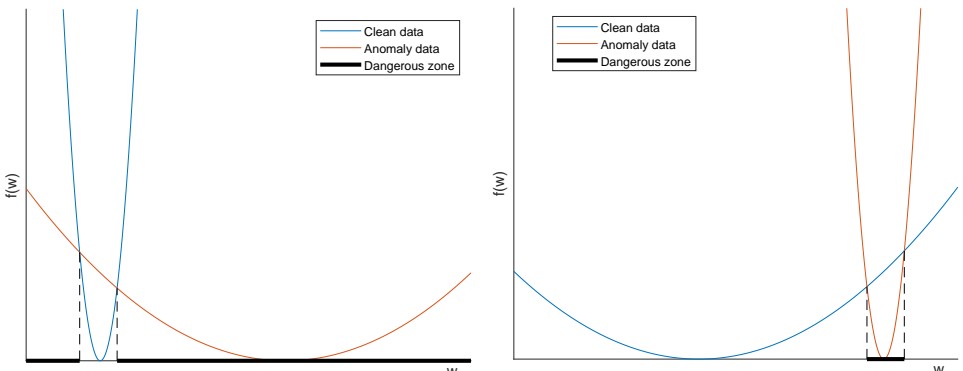

Figure 5: Illustration of the effect of $\kappa$ in theorem 3. The left figure means we have small $L^c_{max}$ and large $\mu^o_{min}$, which leads large $\kappa$. According to our bound, large $\kappa$ is not good in terms of correctness guarantee. As we shall see in the left figure, in this case, the dangerous zone is very large compared to the right figure, where we have large $L^c_{max}$, small $\mu^o_{min}$, and a small $\kappa$. In the right figure, we could see that the orange curve will be dropped since the probability of the orange curve is sampled is very small in our method.

## B   PROBABILITY OF SAMPLING WITH REPLACEMENT

In submitted manuscript, we only show the probability of sampling without replacement due to limit of the space. In here, we show the probability of sampling with replacement:

$$p_i(\mathbf{w}) = \begin{cases} \frac{n^k - (n-1)^k}{n^k} & \text{if } i \leq \beta k; \\ \frac{\sum_{j=0}^{\beta k-1} \binom{k}{j}(i-1)^j [(n-i+1)^{k-j} - (n-i)^{k-j}]}{n^k} & \text{otherwise} \end{cases} \tag{11}$$

## C   SUPPLEMENTARY EXPERIMENTAL RESULTS

In this section, we show the supplementary experiment results, which are omitted in submitted manuscript due to the limit of space.

### C.1   NETWORK HYPERPARAMETER FOR TWO-MOON DATA

The network structure for both autoencoders and our method is the same for a fair comparison. The network has one layer encoder and two layer decoder. In the hidden layer, we have a 0.5 dropout ratio, the number of hidden nodes for all layers are set to be 128. All activation function is chosen to be the tanh function. The maximum training epochs are 200. The stopping criterion is the loss of testing data. The ensemble number is set to be 1000 for our method. Training ratio is 60% and testing ratio is 40%. We use Adam optimizer for both method and the initial learning rate is set to be 3e-4. Batchsize for both methods are 128.

### C.2   NETWORK HYPERPARAMETER FOR BENCHMARK DATA

Specifically, we use a 6-layer fully connected autoencoder with 128 hidden nodes in every layer except for the bottleneck layer, which has 10 hidden nodes. We also set the dropout rate to 0.5 for every hidden layer. The deep neural networks are trained using ADAM, with learning rate initialized to 3e-4 and a batchsize of 128. The maximum epochs is set to be 100 with a stopping criterion determined from the minimum reconstruction loss of the test data. The reconstruction loss function for the opt-digits dataset is cross entropy loss since the feature of this data are all discrete. The rest data reconstruction loss are all mean square error loss. The activation function is chose to be LeakyReLU with $\alpha = 0.1$. For the SVDD, we pretrain the autoencoder for 50 epochs, and use the encoder as the initialization of SVDD except the last layer.

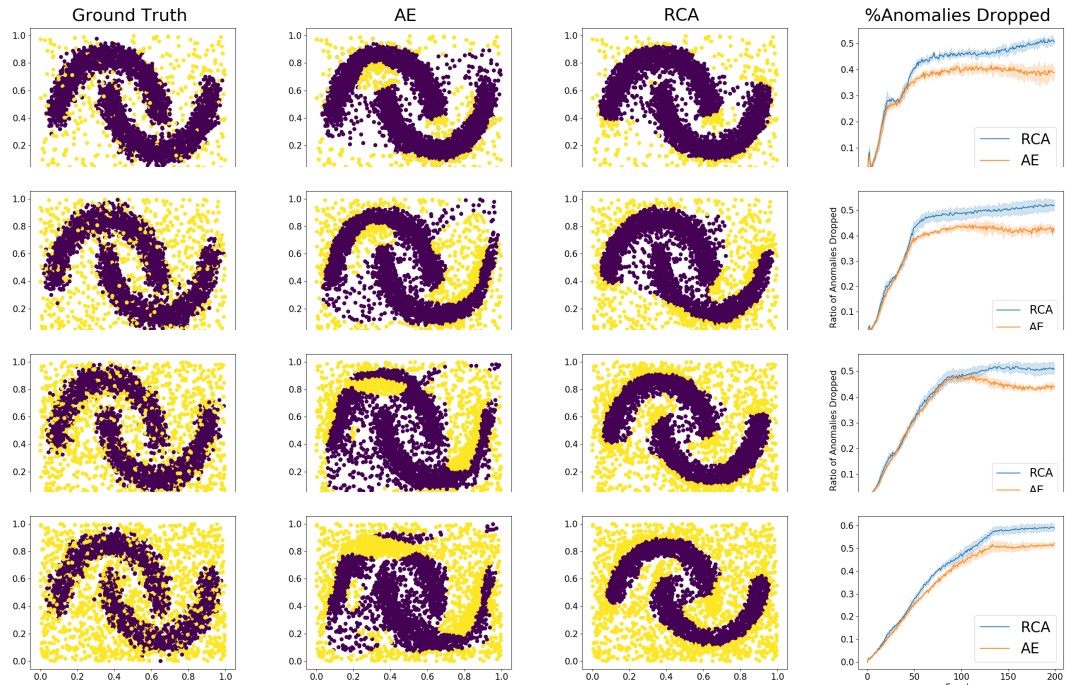

Figure 6: The rows are the results for 10%,20%,30% and 40% anomaly ratio from top to bottom, respectively. The last column shows the fraction of points with highest reconstruction loss that are true anomalies.

## C.3 RESULTS FOR TWO MOON

In submitted manuscript, due to the limit of the space, we only show the results for 10% and 40% anomaly ratio. In this section, we provide the results for 10%, 20%, 30%, and 40% in Figure 6.

