# OpenReview forum: "Unsupervised Anomaly Detection by Robust Collaborative Autoencoders"
_ICLR.cc/2021/Conference — Reject_

### Official Review · AnonReviewer1 · 2020-10-25
**Weak contribution**

**Rating:** 3
**Confidence:** 3

**Review:**

This paper presents a Robust Collaborative Autoencoder (RCA) for unsupervised anomaly detection. The authors focused on the overparameterization of existing NN-based unsupervised anomaly detection methods, and the proposed method aims to overcome the overparameterization problem. The main contibutinos of the proposed method are that (1) it uses two autoencoders, each of which is trained using only selected data points and (2) monte carlo (MC) dropout was used for inference.

Although this paper has an interesting idea, i have doubt about the contributions. My comments are as below.

1) First of all, to me it was very difficult to read this paper. The notations are very confusing.

2) In Introduction section, it is confusing what the main focus of this paper is. They mentioned like "unlike previous studies, our goal is tho learn the weights in an unsupervised learning fashion". But because it seems the topic of this paper belongs to "unsupervised anomaly detection" (the labels indicating whether anomaly or not are assumed available in the training data), the point that your method is in an unsupervised learning fashion is pretty obvious.  You don't need to discuss about "supervised approachs" throughout the paper, but please clearly mention that at the beginning of the introduction section, and only discuss your method and other "unsupervised" anomaly detection methods.

3) If the overparameterization is the problem when we build a NN for unsupervised anomaly segmentation (e.g. autoencoder-AE), we can simply think about various well-known NN regulaization techniques for the AE as remedy. I also think the two parts of the proposed method (corresponding to the contrbutions (1) and (2) )work as regularization for the AE. I'm curious if there's any reason to prefer the proposed method to other regularization techniques?

4) The proposed RCA method involves an ensemble prediction by using MC-dropout (described in section 3.2). The authors metioned this is one of their research contribution, but the use of MC-dropout is quite general in neural network research. Also, while the proposed method definitely benifits from the use of MC-dropout, other unsupervised anomaly detections based on neural networks (e.g. AE, VAE,...) can also improve by employing MC-dropout. The ablation study in Table 1 showed that RCA significantly outperforms RCA-E (RCA without ensembling).  The authors can implement MC-dropout-based ensemble versions of AE, VAE, and other nn-based methods like Deep SVDD, and check whether the proposed only benefits from MC-dropout or RCA just outperforms others regardless of MC-dropout.

---

> ### Author Response · Authors · 2020-11-16
> **Response to Reviewer 1**
>
> a). If the overparameterization is the problem when we build a NN for unsupervised anomaly segmentation (e.g. autoencoder-AE), we can simply think about various well-known NN regularization techniques for the AE as a remedy. I also think the two parts of the proposed method (corresponding to the contributions (1) and (2) )work as regularization for the AE. I'm curious if there's any reason to prefer the proposed method to other regularization techniques?
>
> Our baselines for AE contain several regularizations such as dropout and early stopping. The only difference between AE and RCA is the sample selection and ensemble evaluation. We would appreciate it if the reviewer can provide some references to the alternative regularization and non-regularization techniques that can be used for our experiments.
>
> b). MC-dropout is not new, and should also be used for baselines.
>
> Firstly, we did not see this dropout inference mechanism being used in anomaly detection papers. We understand that the dropout inference has been proposed in other communities, but this is the first time it was applied to anomaly detection settings considering the fact that ensemble training of neural networks can be very expensive.

---

### Official Review · AnonReviewer2 · 2020-10-26
**Paper has poor presentation of results and is a clear reject.**

**Rating:** 3
**Confidence:** 4

**Review:**

The proposed approach differs from autoencoder based anomaly detection approach in the following ways
(a) Autoencoders are trained using only selected data points with small reconstruction errors. These are selected using a sampling scheme with theoretical guarantees on convergence.  The selected points are then shuffled between two autoencoders.

(b) During the testing phase, each autoencoder applies dropout to generate multiple predictions. The averaged ensemble output is used as the final anomaly score.

Some of the issues with this paper

(a) One key issue is why just two autoencoders (which the authors delegate for future work). However, it is key to understanding utility of such an ensemble based shuffling framework.

(b) Poor presentation of results
    1) Figure 2 legend issue
    2) Figure 3 (a) is better presented as a table (Table 3 in Appendix should be here instead). Very hard to interpret it in the current form. Similar comments for Figure 3(b) and Table 1. Also for anomaly detection benchmarking AUC is not sufficient and the authors have to present AUPR or F-1 scores also. I suggest looking at these recent papers for presentation of experimental results

http://proceedings.mlr.press/v108/kim20c/kim20c.pdf


https://proceedings.icml.cc/paper/2020/file/0d59701b3474225fca5563e015965886-Paper.pdf (Goyal et al. ICML 2020)

(c) Theorem 3 might have a connection with the notion of r-robustness presented in https://arxiv.org/abs/2007.07365
      so authors would want to make it clear how they differ.

---

> ### Author Response · Authors · 2020-11-16
> **Response to Reviewer 2**
>
> a). One key issue is why just two autoencoders (which the authors delegate for future work). However, it is key to understanding the utility of such an ensemble-based shuffling framework.
>
> Please refer to our response above in part (c) for reviewer 4.
>
>
> b). The table should be in the main paper instead of in the appendix. AUC score is not enough.
>
> We use a figure instead of a table in the main paper due to page limitations. Also, it is hard to decide whether we should put a figure or a table (i.e. reviewer 4 prefer using a figure). F1-score requires specifying a threshold on the model output, while AUC is more suitable if we want to identify the top-k anomalies. We will add F1-score and AUPRC scores in the future.
>
> c). Relationship to R-robustness
>
> We did not see the similarity between theorem 3 and R-robustness. We would appreciate it if the reviewer can point out more specifically this point.

---

> > ### Comment · AnonReviewer2 · 2020-11-24
> > **Further clarification from Reviewer 2**
> >
> > a) I don't see how your response in part (c) for reviewer 4 helps in addressing my concern. Within the large space of methods in the AD space, I feel your proposal is a bit ad-hoc, and for it to be useful practically, this insight on how the framework performs with multiple AE's is needed and not only for scalability concerns.  I personally feel with multiple AE's your performance might worsen unless you develop a principled approach.
> >
> > b) This is a core presentation issue which severely affects the readability of this paper. Also anomaly detection (AD) experiments need to be exhaustive. In industrial domain nobody uses AUC for evaluating AD performance, it is more often adjusted precision or recall which makes these numbers very critical to include in your main draft.
> >
> > c) My reference was to elicit a response from you on your thoughts on how your definition on top of page 6 before Theorem 3 is different from what has been proposed for a ball of radius r in Camuto et al . 2020. I found a good degree of similarity on how the definition has been used although they do it primarily for VAE's.  However, to clarify further, this comment was not meant to dispute the novelty of your work.

---

### Official Review · AnonReviewer3 · 2020-10-26
**A very interesting idea, with too little evaluation**

**Rating:** 4
**Confidence:** 3

**Review:**

# Summary

The submission tackles unsupervised anomaly detection, specifically in a scenario where supervision labels are not available, only information about the ratio of anomalous examples in the data set. They suggest an architecture consisting of two auto-encoders collaboratively determining anomalous samples and updating their weights based on data that is deemed normal. The authors provide a theoretical analysis of the selection process, and validate anomaly detection performance on a range of experiments.

# Pros

**Interesting challenge**

Tackling the problem of potentially contaminated data heads on instead of side-stepping it with assumptions like guaranteed normalcy of the training data set is an interesting challenge.

**Relative simplicity of the approach**

The suggested algorithm is a remarkably simple extension (or self-regularization?) to vanilla auto-encoding. The changes are fairly minimal, losses and AE architectures remain the same. At the same time, the suggested duplication of AEs and selection of data directly address contaminated anomaly detection data sets. This allows for potentially widespread applicability of the idea to other kinds of data, problems or architectures.

**Theoretical underpinning of the algorithm.**

I applaud the author's effort in examining and motivating the suggested changes not just by experimental results, but by a more rigorous theoretical analysis. This is a big plus in a typically very evaluation- and application driven field. This especially holds true given how small the architectural changes are: proving their legitimacy both theoretically and experimentally can make for a strong contribution.

# Cons

**The motivation and context could be stated more clearly.**

The setting that the authors assume and the contributions to that setting are muddied throughout the paper. Abstract and introduction discuss various problems of DNN-based AD methods, for instance overparameterization. At the core of their setting, however, is contaminated data without any label information beyond (an estimate of) the ratio of anomalies.

I believe the authors should emphasize the setting much clearer, motivate their choice of setting compared to more common approaches in the literature like unsupervised learning on "guaranteed" normal data.

**The theoretical analysis is relatively overemphasized.**

As stated above, I believe the theoretical analysis is a strength of the paper. Spending four pages on the methods section, and two of those an the assumptions, theorems, and remarks, compared to a total of two pages of evaluation, the authors clearly emphasize this aspect of their contribution. From this perspective, I believe the theoretical analysis takes up too much space, especially considering the very application-driven nature of anomaly detection methods. Put bluntly: The theorems are certainly interesting and worth having which is why I consider them a pro; but they are not strong enough to justify the amount of space compared to, e.g., evaluation.

I believe the main text should stick to the theorems, and spend less time on technical details and more time contextualizing the results: How strong are the results? How valid are the assumptions? After all, the proof largely hinges on the assumptions, so they deserve more scrutiny than they currently get. Are the theoretical results reflected in the experimental evaluation? If not, why?

The bit about the integer program to determine the selected data points, taking up half a page, seems like a retrospective justification for the perfectly valid design decision to pick the fraction of examples with lowest reconstruction error, but otherwise does not add much.

**The evaluation is too coarse.**

The previous point dovetails with my main criticism: the evaluation. This should have been a much stronger focus of the paper, given that anomaly detection is very application-driven.

The synthetic data set is nice to examine qualitative results. However, the evaluation is purely qualitative, where quantitative metrics would also be in order. The right-most column of figure 2 is the only hint at performance. We see that a lot of anomalies are not selected in both rows. I might be misunderstanding something, but this hints towards a massive amount of false positives and negatives?

The real-world experiments are lacking a lot of evaluations in my opinion. The analysis is reduced to "winning" the AUC score against the baselines on as many data sets as possible. While that's desirable, it's not helpful in understanding the pros and cons of certain algorithms. This is particularly true given that the experimental setup of contaminated data violates the assumptions of many of the baselines.

Overestimating the share of anomalies is studied, although the given results are very hard to parse. This begs the question: What happens if I underestimate the contamination? This should lead to more anomalies being part of the data used for backprop. Given that the algorithm is based around that ratio, or an estimate thereof, I would have liked to see a stronger focus on it.

As you hypothesize, your selection method biases the representation learning. This should be examined in an experimental evaluation. This is particularly true given the venue.

**The presentation can be improved.**

This is not a decisive point, but I believe potential readers would greatly benefit from improvements in structure, writing, and layout. I have gathered a number of suggestions further down.

# Recommendation

Generally, I believe the suggested architecture and algorithm are worth pursuing and eventually publishing. This may be in contrast with the length of the positive feedback vs. the negative feedback, but in this case the opposite is true: the core idea is intriguing, but the paper on it can be improved.

I believe the paper needs to be more precise and nuanced in answering a potential user's question: When and why should I consider this algorithm? In my view, the paper can and should be improved on two fronts:

1. The experimental evaluation needs to be more thorough, and in particular less focused on "winning" over baselines, but on understanding and showcasing defining properties of the suggested algorithm.
2. The presentation can be made much approachable.

Overall, I believe the paper as is should be rejected. I nevertheless strongly encourage the authors to improve their evaluation and manuscript and resubmit!

# Questions

Generally, the authors aim at fairness by fixing auto-encoder structures. Does this mean that RCAs generally have a multiple of learnable parameters compared to, e.g., the AE baseline? Further, how did you determine the hyperparameters? I would argue an HPS is due when comparing such different models.

Given that VAEs seem to be the baseline that compares most favorably according to your result, it would seem fairly obvious to try "RCVAEs", where everything is the same except reconstruction losses are replaced either by the likelihood term of VAEs or the ELBO directly. Have you considered this, and if so, why haven't you tried it?

Could you elaborate more on how a user with entirely unlabeled data would go about making a good guess for the ratio of anomalies, so as to be able to use your algorithm?

# Further Feedback

I believe the presentation of the method and results could be improved in several places. Take these as suggestions for a revision---I don't see a particular need to address these points in a rebuttal.

The introduction is very long. It contains a substantial amount of related work, and a fairly detailed description of the proposed method. I would encourage the authors to move those bits to the respective dedicated sections and give the reader a more precise problem formulation, and particularly what part of the problems are tackled by the contributions of this submission.

Figure 1 is not particularly illustrative: One can see that subsampling and shuffling is going on, otherwise one has to have a pretty good idea of the method to understand the illustration. On a side-note, I would encourage the authors to investigate tikz or similar alternatives for a cleaner style that is more integrated with the notation of the paper.

Figure 2 cuts the lower part of the top row including a (redundant) legend.

Algorithms 1 and 2 could also be clearer. What is S, \hat{S}_1 etc? First it's a minibatch, then it's the output of an auto-encoder? In algorithm 2, the notation of the forward step changes. \xi is a set, then the last step of the for loop does both set and arithmetic operations on \xi_1 and \xi_2. In this light, I would argue that a more mathematical notation in favor of a programmatic notation would help the reader. This would also shorten the lines and make them more readable.

Generally, the authors aim at fairness by fixing auto-encoder structures. Does this mean that Generally, notation could be a little clearer. \mathcal O is the set of anomalies, which is usually big-O notation, for which you use \mathbb O. Sets sometimes have upper-case Greek letters, sometimes lower-case Greek letters. The probability probability p_i(w) is not actually the probability of w, but the probability of i as a function of w. Such small inaccuracies amount to an unnecessary increased mental burden for the reader.

The results of section 4.2 could be presented in a much clearer format. The figures are illegible at 100% zoom. Putting this aside, both figures 3a and 3b are unnecessarily difficult to process. Consider 3b: The interesting aspect of the figure is the disparity between the lines along certain circle segments. 90% of the graph are uninformative space, and the nuances are lost.

I have a strong, possibly subjective or biased opinion about the way of presentation of table 1. There is something to be said about compressing large tables like e.g. tables 3 and 4 in the appendix into digestible formats. That said, I think wins, draws, and losses are the wrong mind set to approach baselines to begin with. In this particular case it oversimplifies the matter by quite a margin.

The appendix leaves room for improvement, in particular w.r.t. layout and typesetting. Equations should be broken to stay within the margins. Punctuation should conclude equation blocks, and within the environment instead of the next line. Equations should be indented. Delimiters like parentheses should have appropriate height, for instance by making more liberal use of \left and \right for parentheses for better readability.

---

> ### Author Response · Authors · 2020-11-16
> **Response to Reviewer 3**
>
> We thank the reviewer for his/her review and comments to improve the paper.
>
> a). Motivation and context could be stated more clearly. It is better to emphasize that our training data includes contamination.
>
> Thank you for your suggestion. We will try to clarify this in our revision and emphasize more about our unsupervised anomaly detection setting with contaminated training data.
>
> b). The theoretical analysis takes too much space compared to the experiment. Missing empirical evaluation on the theorem.
>
> Thank you for your suggestion. We will consider moving some of the theories to the appendix. Empirical validation of the theorem is difficult since the true conditions (e.g., Lipschitz constant and bound of the objective function) is harder to determine. Also, for deep neural networks, due to the highly non-convex setting, even a change in the random seed can alter the empirical results. Instead, we present rigorous proof in the appendix to show the correctness of the theorem.
>
> Nevertheless, although we did not explicitly provide empirical results on the theorems, there are some connections in the experiment section to our theory. For example, theorem 2 suggests that by discarding data points during training, our method should converge closer to the solution using clean training data. This explains why our method is doing better than AE since we can alleviate the ill-effects brought upon by the anomalies.
>
> c). The right-most column of figure 2 is the only hint at the performance. We see that a lot of anomalies are not selected in both rows.
>
> For the synthetic data, the rightmost column shows that we cannot pick every anomaly data. This is because, in our synthetic data, there are some anomalies hidden in the normal data, which is a challenging setting. For example, if you look at the left-most column, which is the ground truth, you will see that many anomalies also fall in the two-moon manifold. For those anomalies, It is really hard to claim that there is a method that could distinguish those anomalies without strong assumptions. We did not make those anomalies 100% well separated from the normal data in synthetic experiments, this is because that we would like to show even in this challenging scenario, our algorithm can still pick out most of the anomalies.
>
> (d) real-world experiment lacks evaluation.
>
> The win-loss table is for the sake of saving space considering the page limit. Otherwise, there would be too many tables. Also, we agree that many of our baselines assume the contamination does not exist in the training data. It is hard to find strong pure unsupervised anomaly detection baselines for common data since most recent work assumes the clean training data. There are some strong unsupervised anomaly detection baselines in terms of image data, which is not fair to be compared here. Since most of them use self-supervised learning techniques to perform rotation, flipping operations, which are not available for common data.
>
> (d) Does this mean that RCAs generally have a multiple of learnable parameters compared to, e.g., the AE baseline? Further, how did you determine the hyperparameters?
>
> RCAs have twice the number of learnable parameters compared to AE since we use two networks (The two AE without exchanging data results are RCA-SS). The hyperparameters are all the same for AE based results. We use pretty standard hyperparameters in our study (i.e. 3e-4 learning rate, default momentum, early stopping by validation error). Since in an unsupervised setting, there is no available ground truth for tuning our hyperparameters, most of our hyperparameters are the default ones in PyTorch. The only exception is the contamination ratio, which for all baselines, uses the ground truth contamination ratio.
>
> (e) Given that VAEs seem to be the baseline that compares most favorably according to your result, it would seem fairly obvious to try "RCVAEs".
>
> Thank you for your suggestion. We will try to add RCVAEs in the future. In our theory, we did not assume that the loss function should be exactly reconstruction loss, thus at least in theory, changing AE backbone to VAE backbone is also valid. We did not try RCVAEs mainly because AE is more popular than VAE in terms of anomaly detection, we only use AE as our backbone network structure.
>
> (f) Could you elaborate more on how a user with entirely unlabeled data would go about making a good guess for the ratio of anomalies, so as to be able to use your algorithm?
>
>  In many practical applications, users often specify the top-k anomalies they are willing to manually inspect and validate. The ratio of k to the total number of points can be used as an initial estimate of the anomaly ratio. If the ratio is too low (e.g., too many anomalies in the data), the user can continue to increase the ratio until the false positives become too large.

---

> > ### Comment · AnonReviewer3 · 2020-11-23
> > **Response**
> >
> > Thank you for your detailed response. I have some more comments that might be helpful for a revision of your work.
> >
> > Re b: I am aware that it's hard to verify the assumptions. That begs the question: what's the point of the theorem, if I cannot verify its relevance in practice? So your job is to devise experiments that probe this. This is done insufficiently for how much space the theorem covers.
> >
> > Re c: I understood the anomalous data is not clearly separated. The figure is still hard to read. Is 60% good? What should I expect?
> >
> > Re the lower d: If your model has twice as many learnable parameters than the baseline, that's not really a fair comparison. You are regularizing an auto-encoder and claim this works better, so your baseline needs to be as close as possible to your model, in particular in terms of capacity. I think this constitutes a real miss in terms of
> >
> > Re e: AEs have been around much longer, I'd argue VAEs are already quite popular for AD given they have been around for a fraction of the time. Give it a shot!
> >
> > Re f: I'm still unsure what the top-k anomalies are. That seems a bit like a chicken-and-egg problem to me.
> >
> > I will keep my score, given the vote is already rather unanimous and unlikely to shift radically.

---

### Official Review · AnonReviewer4 · 2020-10-27
**Interesting theoretical results, experimental section can be improved**

**Rating:** 4
**Confidence:** 4

**Review:**

### Summary

This manuscript proposes a novel learning method to improve the robustness of unsupervised anomaly detection called robust collaborative autoencoders (RCA). This combines two autoencoders which exchange samples from heterogeneous batches according to the rankings that each individual model assigned.

While the manuscript contains a number of interesting theoretical motivations, its experimental section contains some weakness, and I would hope for it to include a larger set of competitors, as well as a more flexible model class with which RCA is paired.

### Strengths

The paper follows a standard structure and is logically organized. Theoretical results are provided that underpin the sample selection criteria that is used in RCA, in particular. The experimental section compares RCA against some other (however mostly simple) anomaly detection methods, and the authors propose an interesting idea meant to enhance robustness, a particularly desirable property when dealing with anomaly detection.

The authors provide code, which is always a plus. Experimental results are computed from ten random seeds, with standard deviations included. A methodological description is included in Section 3.

### Weaknesses

My main criticism revolves around the extent of the experimental section. Given the generality of AE architectures and their wide applicability to all types of AD (on images, text, etc.), it would have been interesting to learn how RCA fares in different scenarios:  for instance, RCA incorporated with more recent convolutional autoencoding setups is missing in the evaluation. Unfortunately, RCA is evaluated on mostly low-dimensional data, and against simple competitor models/AE architectures.

To counter any doubts around the feasibility of RCA to scale to more complex AE setups and datasets, it would be interesting to see RCA evaluated on more standard real-world benchmark data (such as CIFAR-10, which is widely used in the standard anomaly detection literature), or with more complex autoencoders, e.g. those proposed in Huang et al. (2019). An additional question that remains unexplored is how well RCA can scale to more than two sub-AE modules, and whether this would be a practical thing to do. Experiments (or some discussion) in this direction would be very interesting!

### Additional remarks

* Axes in Figure 2 are illegible.
* The table formatting is sub-optimal, c.f. the instructions for submission.
* The formatting of Alg. 1 and 2 can be improved.
* Table 3 is hard to read, this might be better in a figure.
* Why not move Assumption 5 in the vicinity of Theorem 3, since this is only required there?

---

> ### Author Response · Authors · 2020-11-16
> **Response to Reviewer 4**
>
> We thank the reviewer for his/her review and comments to improve the paper.
>
> a). It would be interesting to see RCA evaluated on more standard real-world benchmark data (such as CIFAR-10).
>
> Note that we did include image data from CIFAR-10 (which have 4096 features extracted using VGG19) in our experiments. However, due to space limitations, we left the results for CIFAR-10 in the appendix section, where we had provided more detailed comparison against other deep learning baselines. We are willing to move these results back into the main paper, if necessary.  We chose to report the results for the ODDS benchmark in the main paper since it was commonly used as an anomaly detection benchmark in many previous studies. Furthermore, the benchmark has diverse types of real-world data from different application domains.
>
> b). It would be interesting to see RCA evaluated on more fancy network structures.
>
> We agree that it would be interesting to apply RCA to more sophisticated network structures such as CNN and LSTM. However, we would like to show that even a simple architecture can benefit from our anomaly detection approach. In fact, some might argue the improved performance could be due to the result of using more complex architecture. Furthermore, given the unsupervised nature of our problem, tuning the hyperparameters for more complex network structure is also trickier.
>
> c). An additional question that remains unexplored is how well RCA can scale to more than two sub-AE modules
>
> We chose to start with two networks first, partly motivated by [1]. We do agree that using multiple networks will be very interesting though it will require further theoretical analysis and more compute-intensive experiments (e.g., training multiple networks would require huge memory for the GPU card). By showing that RCA works even with two networks, this lays the ground-work for us to explore the case for multiple networks in our future research.
>
> d) Table 3 is hard to read, this might be better in a figure.
>
> The figure of table 3 is in the paper (see figure 3(a)).
>
> It is hard to decide on putting a table or a figure since different people have different tastes (i.e. reviewer 2 suggests putting the table in the main text).
>
>
>
> [1] Han et al.(2018). Co-teaching: Robust training of deep neural networks with extremely noisy labels. In Advances in neural information processing systems (pp. 8527-8537).

---

### Decision · Program_Chairs · 2021-01-07
**Final Decision**

**Decision:**

Reject

**Comment:**

The paper describes an autoencoder-based approach to anomaly detection.  The main weakness—not untypical for papers in this application area—is the experimental section.  The problem itself may be not well-defined, and of course that makes practical comparison difficult. Perhaps different measures—e.g., remaining life—may be better to compare on, and give better data sets.